# Continuous Q-Score Matching: Diffusion Guided Reinforcement Learning for Continuous-Time Control

Chengxiu Hua[1], Jiawen Gu[1*], Yushun Tang[1,2*]

[1]Southern University of Science and Technology, [2]Huawei Technologies Co., Ltd.
`12331005@mail.sustech.edu.cn, jwgu.hku@outlook.com, tangys2022@mail.sustech.edu.cn`

## Abstract

Reinforcement learning (RL) has achieved significant success across a wide range of domains, however, most existing methods are formulated in discrete time. In this work, we introduce a novel RL method for continuous-time control, where stochastic differential equations govern state-action dynamics. Departing from traditional value function-based approaches, our key contribution is the characterization of continuous-time Q-functions via a martingale condition and the linking of diffusion policy scores to the action gradient of a learned continuous Q-function by the dynamic programming principle. This insight motivates Continuous Q-Score Matching (CQSM), a score-based policy improvement algorithm. Notably, our method addresses a long-standing challenge in continuous-time RL: preserving the action-evaluation capability of Q-functions without relying on time discretization. We further provide theoretical closed-form solutions for linear-quadratic (LQ) control problems within our framework. Numerical results in simulated environments demonstrate the effectiveness of our proposed method and compare it to popular baselines.

## 1 Introduction

RL has achieved substantial success across a wide range of domains over the past decade [44]. Most existing approaches adopt a discrete-time formulation, typically modeled as a Markov Decision Process (MDP) [36, 16], where agents interact with the environment at fixed time intervals. However, many real-world systems—such as autonomous driving in dynamic traffic conditions [47], robotic manipulation [34], and high-frequency algorithmic trading [27]—exhibit continuous, fine-grained dynamics that are inadequately captured by discrete-time models. These applications naturally motivate the need for continuous-time reinforcement learning (CT-RL) frameworks that more faithfully represent the temporal structure of decision-making. Recent works on CT-RL have explored stochastic modeling using stochastic differential equations (SDEs) [11, 21], entropy-regularized exploration techniques [15], and model-free learning methods for diffusion generative models fine-tuning [58, 17] and financial applications [19, 4].

Despite these advances, value-based methods like Q-learning [56]—a cornerstone of discrete-time RL—remain challenging to adapt to the continuous-time setting. Traditional Q-learning algorithms (e.g., SARSA [44], DQN [32]) rely on estimating state-action value functions via temporal-difference (TD) learning and have demonstrated strong performance in discrete action spaces. There is a line of work on discretizing continuous action spaces to apply Q-learning in high-dimensional continuous control settings [49, 40, 20]. Discretizing continuous actions is a common approach to extend Q-learning, but it often struggles with scalability in high-dimensional spaces and relies on

---

* Corresponding authors.

discrete-time assumptions. However, extending Q-learning to continuous action spaces introduces major challenges. Early work [13] proposed neural network-based formulations for continuous Q-learning, but later studies [8] reported severe performance drops in high-dimensional settings due to the curse of dimensionality. Moreover, when Q-learning is directly extended to continuous time, the Q-function tends to collapse into an action-independent value function [46], losing its ability to distinguish between actions—a critical property for decision-making. In [25], they show that the discrete Q-learning algorithm is noisier and slower in convergence speed compared with their proposed continuous PG and little q learning algorithms.

To bridge this gap, recent work has explored Q-function-based frameworks that preserve action dependence in continuous time. For example, [10] replaced the Q-function with a generalized Hamiltonian, while [25] proposed little q-learning, a time-discretization-free method based on first-order Q-function approximations, which achieved faster convergence than discrete-time soft Q-learning (SARSA). Score-matching-based methods [35] have also emerged as promising tools for learning diffusion policies, though they still rely on time discretization. Other approaches, such as [26], incorporate the action as a state variable by constraining the action process to be absolutely continuous with bounded growth. However, these methods are limited to deterministic dynamics and require upfront discretization of the continuous-time problem. These constraints underscore a key open challenge: how to design a principled and scalable Q-learning framework for stochastic, continuous-time environments that maintains the action characteristics of the Q-function. We address this by introducing a novel Q-function formulation and a corresponding algorithm, Continuous Q-Score Matching (CQSM), which operates directly in continuous time and supports stochastic dynamics.

**Major contributions.** This work makes the following key contributions:
(1) **Continuous-Time Q-Function Characterization.** We derive a Bellman equation (also known as the Feynman-Kac formula) for continuous-time Q-functions. This bridges discrete Q-learning with continuous control theory. We rigorously characterize Q-functions for a given score function using a martingale condition defined over an enlarged filtration that incorporates both state and action noise. Based on this foundation, we propose new algorithms for direct Q-function learning, analogous to policy evaluation and policy gradient methods in [23, 24].
(2) **Score Improvement Theorem and CQSM Algorithm.** We establish a score improvement theorem by the dynamic programming principle that enables principled policy updates in continuous time. By coupling the Q-function with the score function of a diffusion policy, we develop a model-free, actor-critic-style algorithm: Continuous Q-Score Matching (CQSM). This algorithm facilitates efficient policy improvement using only the denoising score function.
(3) **Analytical Validation via LQ Control.** We resolve the LQ problem under measurable scores, demonstrating the theoretical soundness of our framework. LQ problems are fundamental in control theory, as they serve as tractable approximations to more complex nonlinear systems, with practical relevance in areas such as algorithmic trading [2] and resource management [12]. Using analytical solutions, we compare CQSM to policy gradient and little q-learning methods, which are the action-independent value-based RL models, highlighting both its theoretical guarantees and numerical advantages.

In Section 2, we review related work relevant to our method. Section 3 introduces the continuous-time reinforcement learning formulation using stochastic differential equations and presents key preliminary results. In Section 4, we develop the Q-learning theory in continuous time, establishing martingale characterizations. We further extend the analysis to the infinite-horizon setting and prove the score improvement theorem. Section 5 presents numerical evaluations on LQ control tasks, comparing CQSM with policy gradient and little q-learning methods. Finally, Section 6 concludes with a summary and discussion of future directions.

## 2 Related Works

In this section, we review existing work across four areas related to our method: stochastic optimal control, continuous reinforcement learning, diffusion Q-learning, and behavior cloning.

**Stochastic Optimal Control.** Our approach builds on classical stochastic optimal control theory [57], particularly through the development of the Hamilton–Jacobi–Bellman (HJB) equation for

continuous-time Q-functions. While traditional stochastic control frameworks often assume full knowledge of the system dynamics [3], we instead assume a model for the dynamics of state-action pairs. This shift motivates new theoretical developments that underpin our Q-learning framework.

**Continuous Reinforcement Learning.** The formulation of CT-RL in stochastic settings—where state evolution follows a stochastic differential equation (SDE)—dates back to [33, 7], though early work lacked data-driven learning mechanisms. Recent advances have introduced more practical formulations. For instance, [52] proposed an exploratory control framework for continuous RL, while [53], [23], and [24] extended this line of work to mean-variance objectives, policy evaluation, and policy gradients, respectively. [25] further introduced the notion of a little q-value, leading to a continuous analogue of Q-learning. Additional developments include mean-field RL with continuous dynamics [28], jump-diffusion extensions [14], and infinite-horizon variants of TRPO and PPO [59]. Building on these foundations, we advance the study of continuous-time Q-learning under diffusion policies.

**Diffusion Q-Learning.** Diffusion Q-learning [55] integrates diffusion models with Q-learning by using Q-values as training objectives and backpropagates through the diffusion model. More recently, [5] proposed a model-free online RL method based on diffusion policies. [35] further established a connection between diffusion-based policies and the Q-function by relating the policy score to the action gradient of the Q-function. Building on this line of work, [54] employed entropy estimation to balance exploration and exploitation in diffusion policies, improving the performance of the policy. In parallel, [30] generalized diffusion model training by reweighting the conventional denoising score matching loss, leading to two efficient algorithms for training diffusion policies in online RL without requiring samples from optimal policies. However, their approach relies on time discretization and requires injecting noise into the final action of the diffusion chain. In contrast, our work preserves the full action-evaluation capability of the Q-function in continuous time, without relying on any discretization in either time or action space. Our method is grounded in a martingale-based formulation of the HJB equation, which provides a principled theoretical foundation for continuous-time Q-learning.

**Behavior Cloning.** Behavior cloning focuses on imitating expert trajectories without access to reward signals. Diffusion models are particularly well-suited for this task due to their generative flexibility and natural alignment with score-matching objectives. Recent works [22, 37] have applied diffusion models to behavior cloning by framing policy learning as a distribution-matching problem over expert data. These approaches inspired our incorporation of score-matching terms into the objective. However, our framework goes beyond imitation, enabling policy improvement through Q-function learning in continuous time.

## 3 Formulation and Preliminaries

In this section, we introduce the continuous-time RL formulation using stochastic differential equations and present key preliminary results.

**Notation.** We introduce the non-standard notation used throughout the main text and appendix. For a vector $x$, denote by $\|x\|_2$ the Euclidean norm of $x$. For a function $f$ on an Euclidean space, $\nabla f$ (resp. $\nabla^2 f$) denotes the gradient (resp. the Hessian) of $f$. The Kullback–Leibler (KL) divergence of two positive density functions $f, g$ is defined as $D_{KL}(f\|g) := \int_A \log \frac{f(a)}{g(a)} f(a)\,\mathrm{d}a$. Define an operator $\mathcal{L} : C^{2,2}(\mathbb{R}^n \times \mathbb{R}^d) \cap C(\mathbb{R}^n \times \mathbb{R}^d) \to C(\mathbb{R})$ [57] associated with the diffusion process as:

$$\mathcal{L}\varphi(x,a) := \nabla_x\varphi(x,a)^\top b_X + \nabla_a\varphi(x,a)^\top \Psi + \frac{1}{2}\mathrm{tr}\left(\sigma_X\sigma_X^\top\nabla_x^2\varphi(x,a)\right) + \frac{1}{2}\mathrm{tr}\left(\sigma_a\sigma_a^\top\nabla_a^2\varphi(x,a)\right).$$

In both the main text and the proofs, we refer frequently to the *score* of the action distribution, denoted by $\Psi$. This vector field defines the temporal evolution of actions and serves as a proxy for the true score $\nabla_a \log \pi(a|x)$ where $\pi$ refers to the action distribution.

**Continuous RL.** Let $d, n$ be positive integers, $T > 0$. We denote the state as $X_t \in \mathbb{R}^n$ and the action as $a_t \in \mathbb{R}^d$ with $t \in [0, T]$. We consider the following stochastic, continuous-time setting for

state and action dynamics:

$$\mathrm{d}X_t = b_X(t, X_t, a_t)\,\mathrm{d}t + \sigma_X(t, X_t, a_t)\,\mathrm{d}B_t^X, \mathrm{d}a_t = \Psi(t, X_t, a_t)\,\mathrm{d}t + \sigma_a(t, X_t, a_t)\,\mathrm{d}B_t^a, \quad (1)$$

where $\Psi : [0, T] \times \mathbb{R}^n \times \mathbb{R}^d \to \mathbb{R}^d$ corresponds to the score of our policy, which serves as the primary optimization variable in this setting, $b_X, b_a : [0, T] \times \mathbb{R}^n \times \mathbb{R}^d \to \mathbb{R}^n$ corresponds to the continuous state and action dynamics, and $\sigma_X, \sigma_a$ are functions from $[0, T] \times \mathbb{R}^n \times \mathbb{R}^d$ to positive semidefinite matrices in $\mathbb{R}^{n \times n}$ and $\mathbb{R}^{d \times d}$ respectively. The processes are driven by two independent Brownian motions: $B^X = \{B_s^X, s \geq 0\}$ and $B^a = \{B_s^a, s \geq 0\}$. All processes are defined on a filtered probability space $(\Omega, \mathcal{F}, \mathbb{P}; \{\mathcal{F}_s\}_{s \geq 0})$ where $\{\mathcal{F}_s\}_{s \geq 0}$ is the natural filtration generated by a standard $n$-dimensional Brownian motion $B^X$ and a standard $d$-dimensional Brownian motion $B^a$. The (continuous-time) Q-function under any given $\Psi$ is defined as

$$Q(t, x, a; \Psi) = \mathbb{E}^{\mathbb{P}} \left[ \int_t^T \left[ r(s, X_s, a_s) - \frac{1}{2}\lambda\|\Psi(s, X_s, a_s)\|_2^2 \right] \mathrm{d}s + h(X_T, a_T) \Big| X_t = x, a_t = a \right],$$
$$(2)$$

where $\mathbb{E}^{\mathbb{P}}$ is the expectation with respect to both Brownian motions $B_t^X$ and $B_t^a$, $r : [0, T] \times \mathbb{R}^n \times \mathbb{R}^d \to \mathbb{R}$ and $h : \mathbb{R}^n \times \mathbb{R}^d \to \mathbb{R}$ are running and lump-sum reward function, respectively, and $\lambda > 0$ is the regularization coefficient governing the cost of large score magnitudes. The goal is to find an optimal score function $\Psi^* \in \Pi$ where $\Pi$ denotes the set of admissible diffusion scores, such that the optimal Q-function

$$Q(t, x, a) = \sup_{\Psi \in \Pi} Q(t, x, a; \Psi). \quad (3)$$

We now give a precise definition of the admissible score set $\Pi$.

**Definition 1.** *A score $\Psi$ is called admissible if*
*(i) $\Psi := \{\Psi(t, X_t, a_t) : t \geq 0\}$ is adapted;*
*(ii) $\mathbb{E}^{\mathbb{P}} \left[ \int_0^T \|\Psi(s, X_s, a_s)\|_2^2\,\mathrm{d}s \right] < \infty.$*

Details regarding the well-posedness of the control problem (1)-(3) are provided in Appendix A.

The score matching term $\frac{1}{2}\lambda\|\Psi(X_s, a_s)\|_2^2$ in the objective can be interpreted from the following two perspectives:
**Quadratic Execution Costs.** Let $a_t$ denote the investor's portfolio position. The score matching term captures the quadratic costs of execution trades of size $\Psi\,\mathrm{d}t$, where $\lambda$ quantifies the level of transaction costs. This is consistent with execution cost models in portfolio optimization [1];
**Policy Regularization via KL Divergence.** Suppose the diffusion coefficient $\sigma_a$ is deterministic and $\Psi(t, X_t, a_t) = b_a(t, X_t, a_t) + u(t, X_t, a_t)\sigma_a(t)$ where $u$ represents a control. Under this setup, the score-matching cost admits an interpretation as a KL divergence between trajectory distributions. Specifically, let $\pi^{\text{base}}(a_T|x, a)$ denote the distribution over terminal actions when $u = 0$ and let $\pi^u(a_T|x_t, a_t)$ denote the distribution under control $u$. By the Section 3 in [6]), we have

$$D_{KL}(\pi^u(a_T|x, a)\|\pi^{\text{base}}(a_T|x, a)) = \mathbb{E}^{\mathbb{P}} \left[ \int_t^T \frac{1}{2}\|u(s, X_s, a_s)\|_2^2\,\mathrm{d}s \Big| X_t = x, a_t = a \right]. \quad (4)$$

Setting $\lambda = \|\sigma_a(t)\|^2$, the original score-matching term aligns exactly with this KL regularization. The corresponding objective can then be interpreted as:

$$Q(t, x, a) = \sup_{u \in \Pi} \mathbb{E}^{\mathbb{P}} \left[ \int_t^T r(X_s, a_s)\,\mathrm{d}s + h(X_T, a_T) \Big| X_t = x, a_t = a \right]$$
$$- \lambda D_{KL}(\pi^u(a_T|x, a)\|\pi^{\text{base}}(a_T|x, a)). \quad (5)$$

Thus, the KL term encourages the optimal policy to remain close to the base dynamics, introducing a form of regularized policy improvement.

## 4   Continuous Q-Score Matching Algorithm

In this section, we develop a continuous-time Q-learning framework using a martingale characterization and the HJB equation. This offers an alternative approach to policy improvement.

## 4.1 Dynamic Programming and HJB Equation for Q-Function

By the dynamic programming principle, the Q-function satisfies the following HJB equation:

$$\sup_{\Psi \in \Pi} \left\{ \mathcal{L}Q(t,x,a) + \frac{\partial Q}{\partial t}(t,x,a) + r(t,x,a) - \frac{1}{2}\lambda\|\Psi(t,x,a)\|_2^2 \right\} = 0. \tag{6}$$

Note that the terms $\nabla_x Q \cdot b_X(t,x,a), \nabla_a Q \cdot b_a(t,x,a), \frac{1}{2}\text{tr}\left(\sigma_X \sigma_X^\top \nabla_x^2 Q\right), \frac{1}{2}\text{tr}\left(\sigma_a \sigma_a^\top \nabla_a^2 Q\right)$ and $r(t,x,a)$ are all independent of $\Psi$. Hence, the supremum in (6) is attained at $\Psi^*(t,x,a) = \lambda^{-1}\nabla_a Q(t,x,a)$. Substituting this back into the HJB equation gives the following nonlinear partial differential equation characterizing the optimal Q-function:

$$\begin{cases} \dfrac{\partial Q}{\partial t}(t,x,a) + r(t,x,a) + \nabla_x Q(t,x,a)^\top \cdot b_X(t,x,a) + \dfrac{1}{2}\lambda^{-1}\|\nabla_a Q(t,x,a)\|_2^2 \\ + \dfrac{1}{2}\text{tr}\left(\sigma_X(t,x,a)\sigma_X(t,x,a)^\top \nabla_x^2 Q(t,x,a)\right) + \dfrac{1}{2}\text{tr}\left(\sigma_a(t,x,a)\sigma_a(t,x,a)^\top \nabla_a^2 Q(t,x,a)\right) = 0, \\ Q(T,x,a) = h(x,a). \end{cases} \tag{7}$$

We now focus on the optimal Q-function associated with the optimal score $\Psi^*$. To avoid unduly technicalities, we assume throughout this paper that the Q-function $Q \in C^{1,2,2}([0,T) \times \mathbb{R}^n \times \mathbb{R}^d) \cap C([0,T] \times \mathbb{R}^n \times \mathbb{R}^d)$ satisfies the polynomial growth condition in the joint state-action variable $z = (x,a)$. The following theorem establishes the key martingale characterization underpinning policy evaluation for diffusion-based policies.

**Theorem 1.** *If $Q(\cdot,\cdot,\cdot;\Psi)$ is the Q-function associated with the score $\Psi$ if and only if it satisfies terminal condition $Q(T,x,a;\Psi) = h(x,a)$, and for all $(x,a) \in \mathbb{R}^n \times \mathbb{R}^d$, the following process*

$$M_s = Q(s,X_s,a_s;\Psi) + \int_t^s \left[ r(u,X_u,a_u) - \frac{1}{2}\lambda\|\Psi(u,X_u,a_u)\|_2^2 \right] \mathrm{d}u \tag{8}$$

*is a $(\{\mathcal{F}_s\}_{s\geq 0}, \mathbb{P})$-martingale on $[t,T]$. Conversely, if there is a continuous Q-function $\tilde{Q}$ such that for all $(x,a) \in \mathbb{R}^n \times \mathbb{R}^d$, $\tilde{M}_s$ is a martingale, where*

$$\tilde{M}_s = \tilde{Q}(s,X_s,a_s;\Psi) + \int_t^s \left[ r(u,X_u,a_u) - \frac{1}{2}\lambda\|\Psi(u,X_u,a_u)\|_2^2 \right] \mathrm{d}u, \tag{9}$$

*and $\tilde{Q}(T,x,a;\Psi) = h(x,a)$, then $\tilde{Q} = Q$ on $[0,T] \times \mathbb{R}^n \times \mathbb{R}^d$. Furthermore, the martingale property of $M \in L^2_{\mathcal{F}}([0,T])$ is equivalent to the following orthogonality condition:*

$$\mathbb{E}^{\mathbb{P}} \int_0^T \xi_t \left[ \mathrm{d}Q(t,X_t,a_t;\Psi) + r(t,X_t,a_t)\,\mathrm{d}t - \frac{1}{2}\lambda\|\Psi(t,X_t,a_t)\|_2^2\,\mathrm{d}t \right] = 0, \tag{10}$$

*for any test process $\xi \in L^2_{\mathcal{F}}([0,T]; Q(\cdot,X_.,a_.;\Psi))$.*

Proof can be found in Appendix B.1. In summary, the martingality of the process defined in Equation (8) under a given score $\Psi$ is both necessary and sufficient for $Q$ to be the corresponding action value function.

## 4.2 Score Evaluation of the Q-Function

We now discuss how the HJB equation can be used to design a Q-learning algorithm for estimating $Q(x,a;\Psi)$ using sample trajectories. A number of algorithms can be developed based on two types of objectives: to minimize the martingale loss function or to satisfy the martingale orthogonality conditions. Following [23], we leverage the martingale orthogonality condition, which states that for any $T > 0$ and a suitable test process $\xi$,

$$\mathbb{E}^{\mathbb{P}} \int_0^T \xi_t \left\{ \mathrm{d}Q(t,X_t,a_t;\Psi) + \left[ r(t,X_t,a_t) - \frac{1}{2}\lambda\|\Psi(t,X_t,a_t)\|_2^2 \right] \mathrm{d}t \right\} = 0. \tag{11}$$

To approximate the Q-function, we consider a parameterized family $Q^\theta(\cdot,\cdot,\cdot;\Psi)$ where $\theta \in \Theta \subset \mathbb{R}^{L_\theta}$ (in principle, we need at least $L_\theta$ equations as our martingale orthogonality conditions in order to fully

determine $\theta$.) and choose the special test function $\xi_t = \frac{\partial Q^\theta}{\partial \theta}(t, X_t, a_t; \Psi)$. Stochastic approximation [38] leads to the online update:

$$\theta \leftarrow \theta + \alpha_\theta \frac{\partial Q^\theta}{\partial \theta}(t, X_t, a_t; \Psi) \left( \mathrm{d}Q(t, X_t, a_t; \Psi) + \left[ r(t, X_t, a_t) - \frac{1}{2}\lambda \|\Psi(t, X_t, a_t)\|_2^2 \right] \mathrm{d}t \right) \quad (12)$$

where $\alpha_\theta$ is a learning rate. This recovers the mean-squared TD error (MSTDE) method for policy evaluation in the discrete RL [43]. We must, however, stress that testing against this specific function is theoretically not sufficient to guarantee the martingale condition. Additional discussions are provided in Appendix D.

### 4.3 Policy Optimization via Matching the Score to the Q-function

Now we extend the analysis to the infinite-horizon setting. The dynamics of the state and action processes are given by:

$$\mathrm{d}X_t = b_X(X_t, a_t)\,\mathrm{d}t + \sigma_X(X_t, a_t)\,\mathrm{d}B_t^X, \mathrm{d}a_t = \Psi(X_t, a_t)\,\mathrm{d}t + \sigma_a(X_t, a_t)\,\mathrm{d}B_t^a, \quad (13)$$

and the following discounted Q-function under any given $\Psi$:

$$Q(x, a; \Psi) = \mathbb{E}^{\mathbb{P}}\left[ \int_t^{+\infty} e^{-\beta(s-t)} \left[ r(X_s, a_s) - \frac{1}{2}\lambda \|\Psi(X_s, a_s)\|_2^2 \right] \mathrm{d}s \Big| X_t = x, a_t = a \right], \quad (14)$$

where $\beta > 0$ is a discount factor that measures the time-depreciation of the objective value (or the impatience level of the agent) and the optimal Q-function $Q(x, a) = \sup_{\Psi \in \Pi} Q(x, a; \Psi)$.

Note that in this case, the Q-function does not depend on time explicitly. As a result, there is no terminal condition, but instead we have a growth condition $\mathbb{E}^{\mathbb{P}}[e^{-\beta t}Q(X_t, a_t; \Psi)] \to 0$ as $t \to \infty$. Again, using the dynamic programming principle, we have

$$\sup_{\Psi \in \Pi} \left\{ \mathcal{L}Q(x, a) - \beta Q(x, a) + r(x, a) - \frac{\lambda}{2}\|\Psi(x, a)\|_2^2 \right\} = 0. \quad (15)$$

We now introduce an alternative method for updating policies based on Q-function estimates that avoids the use of policy gradients. The following result serves as a score improvement theorem, analogous to the classic policy improvement theorem in reinforcement learning.

**Theorem 2.** *Let $\Psi$ be any given score function and let the associated Q-function $Q(\cdot, \cdot; \Psi) \in C^{2,2}(\mathbb{R}^n \times \mathbb{R}^d) \cap C^0(\mathbb{R}^n \times \mathbb{R}^d)$. Suppose further that the score function $\Psi^1$ defined by $\Psi^1 = \lambda^{-1}\nabla_a Q(x, a; \Psi)$ for some $\lambda > 0$ is admissible. Then $Q(x, a; \Psi^1) \geq Q(x, a; \Psi), (x, a) \in \mathbb{R}^n \times \mathbb{R}^d$.*

Proof can be found in Appendix B.2.

[35] similarly constructed a score function $\Psi^1$, but their approach only determines the direction of the policy update, without specifying the magnitude of the update vector. This result implies that iteratively updating the score function by aligning it with the action gradient of the Q-function leads to monotonic improvement in the Q-values. In other words, setting $\Psi \leftarrow \lambda^{-1}\nabla_a Q(x, a)$ guarantees an improvement in the resulting Q-function globally. Figure 1 shows a visual description of Theorem 2 and the implied policy update direction via CQSM. Building on this theoretical foundation, we now describe how to implement a sample-based update using a parameterized score function $\Psi^v$ with the parameter $v \in \mathbb{R}^{L_v}$. To match the direction of $\nabla_a Q(x, a)$, we define the update as: $v \in \arg\min \frac{1}{2}\|\Psi^v(x, a) - \lambda^{-1}\nabla_a Q(x, a)\|^2$. This yields the CQSM as shown in Algorithm 1.

**Policy sampling.** Consider the dynamics of (13) for a fixed state and setting $\sigma_a(x, a) = \sqrt{2}I_d$. Given certain conditions, under appropriate regularity conditions, the stationary distribution of the action $a_t$ as $t \to \infty$ for any fixed $x \in \mathbb{R}^n$, denoted $\pi(a|x)$: $\pi(a|x) \sim e^{\frac{1}{\lambda}Q(x, a)}$. That is, the stationary action distribution corresponds to a Boltzmann distribution over actions: $\pi(a|x) = \frac{1}{Z}e^{\frac{1}{\lambda}Q(x, a)}$, where $Z = \int_{\mathbb{R}^d} e^{\frac{1}{\lambda}Q(s, a)}\,\mathrm{d}a$. This gives rise to a soft optimal policy, where the action distribution is shaped by the Q-values, rather than relying on the soft Hamiltonian or auxiliary q-functions used in the soft actor-critic literature (e.g., [24, 25]). However, for general $\sigma_a(x, a)$, the stationary distribution $\pi(a|x)$ may not have a closed-form expression [48]. More details about action sampling can be found in Appendix E.

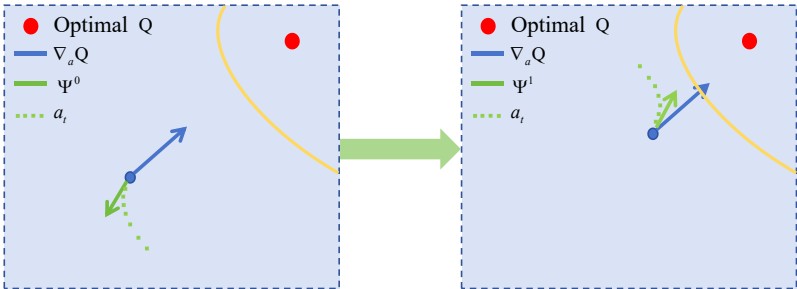

Figure 1: The left image shows a randomly initialized score function $\Psi^0$, and the right shows the updated score $\Psi^1$ after one step. If a discrepancy exists between the score $\Psi$ (green vector) and the action gradient $\nabla_a Q(x, a)$ (blue vector), then aligning $\Psi$ with $\nabla_a Q$ yields a strict improvement in Q-value at $(x, a)$.

Here we highlight several hyperparameters: the trajectory truncation parameter (time horizon) $T$ (needs to be sufficiently large); the total sample size $N$ or the sampling interval $\Delta t$, with $N \cdot \Delta t = T$. We define the observation times as $t_k := k \cdot \Delta t, k = 0, \ldots, N - 1$, at which data is collected from the simulated environment, denoted as Environment$_{\Delta t}$.

---

**Algorithm 1** Continuous Q-Score Matching (CQSM)

---

1: **Inputs:** initial state $x_0$, time step $\Delta t$, initial learning rates $\alpha_\theta, \alpha_v$ and learning rate schedule function $l(\cdot)$ (a function of time), the action value function $Q^\theta(\cdot, \cdot)$, the score function $\Psi^v(\cdot, \cdot)$, the test function $\xi(x_{\cdot \wedge t}, a_{\cdot \wedge t})$, the initial parameter $\theta_0, v_0$, and regularization parameter $\lambda$.
2: **Required:** Environment simulator $(x', r) = $ Environment$_{\Delta t}(x, a)$.
3: **for** $k = 0, \ldots, N - 1$ **do**
4:     Sample action $a$ via iterative denoising: $a^T \sim \mathcal{N}(0, I) \to a^0$, using score $\Psi^v$: $a \sim \pi(x)$
5:     Simulate environment step: $(x', r) = $ Environment$_{\Delta t}(x, a)$. Update state: $x_{t_{k+1}} \leftarrow x'$.
6:     Sample new action $a' \sim \pi(x')$ via denoising with $\Psi^v$. Store $a_{t_{k+1}} \leftarrow a'$
7:     Compute test function: $\xi_{t_k} = \xi(x_{t_0:k}, a_{t_0:k})$
8:     Compute temporal difference:

$$\delta = Q^\theta(x', a') - Q^\theta(x, a) + r(x, a)\Delta t - \frac{\lambda}{2}\|\Psi^v(x, a)\|^2 \Delta t - \beta Q^\theta(x, a)\Delta t$$

9:     Compute parameter updates:

$$\Delta\theta = \xi_{t_k} \cdot \delta, \quad \Delta v = \left(\frac{1}{\lambda}\nabla_a Q^\theta(x, a) - \Psi^v(x, a)\right)\frac{\partial \Psi^v}{\partial v}(x, a)$$

10:     Update parameters:

$$\theta \leftarrow \theta + l(k\Delta t)\alpha_\theta \Delta\theta, \quad v \leftarrow v + l(k\Delta t)\alpha_v \Delta v$$

11:     Set $x \leftarrow x'$
12: **end for**

---

## 5 Experiments

In this section, we present numerical evaluations on LQ control tasks. Additional experimental results are provided in Appendix F. We compare the performance of our proposed CQSM algorithm against continuous time policy gradient [24] and continuous time little q-learning methods [25]. Below, we briefly review these baseline methods.

- CT-RL policy gradient: Given an admissible policy, this method first performs policy evaluation to estimate the corresponding value function. It then computes the policy gradient

as: $g(t, x; \phi) = \frac{\partial}{\partial \phi} J(t, x; \pi^\phi)$ ($J$ is a value function). [24] transforms policy gradient into policy evaluation to develop a policy gradient algorithm.

- CT-RL q-learning: The little q-function is defined as the first-order derivative of the Q-function with respect to $\Delta t$. Policy improvement is achieved via $\pi^\phi(a|t, x) = \frac{\exp\{\frac{1}{\lambda} q^\phi(t,x,a)\}}{\int \exp\{\frac{1}{\lambda} q^\phi(t,x,a)\} \, da}$, leading to a continuous-time q-learning theory.

**Linear-Quadratic Stochastic Control.** We now focus on the family of stochastic control problems with linear state dynamics

$$b_X(x, a) = Ax + Ba \quad \text{and} \quad \sigma_X(x, a) = Cx + Da, \sigma_a(x, a) = \sqrt{2}, x, a \in \mathbb{R}, \qquad (16)$$

where $A, B, C, D \in \mathbb{R}$ and the quadratic reward

$$r(x, a) = -\left(\frac{M}{2}x^2 + Rxa + \frac{N}{2}a^2 + Px + P'a\right), \qquad (17)$$

where $M \geq 0, N > 0, R, P, P' \in \mathbb{R}$. If $D \neq 0$, then one smooth solution to the HJB equation

$$\beta Q(x, a) - Q_x b(x, a) - \frac{1}{2\lambda}Q_a^2 - \frac{1}{2}\sigma_X^2 Q_{xx} - \frac{1}{2}\sigma_a^2 Q_{aa} - r(x, a) = 0, \qquad (18)$$

is given by $Q(x, a) = \frac{1}{2}k_0 x^2 + k_1 x + \frac{1}{2}k_2 a^2 + k_3 a + k_4 xa + k_5$ where

$$\begin{cases} k_0 = \frac{1}{\lambda(\beta - 2A - C^2)}k_4^2 - \frac{M}{\beta - 2A - C^2} \\ k_1 = \frac{1}{\lambda(\beta - A)}k_3 k_4 - \frac{P}{\beta - A} \\ k_2 = \frac{\beta}{2}\lambda - \lambda\sqrt{\frac{\beta^2}{4} + \frac{1}{\lambda}(N - 2\varpi)} \\ k_3 = -\frac{BP + P'(\beta - A)}{(\beta - A)\left(\beta - \frac{B}{\lambda(\beta - A)}k_4 - \frac{1}{\lambda}k_2\right)} \\ k_4 = \frac{-\lambda(\beta - 2A - C^2)B \pm \sqrt{\lambda^2(\beta - 2A - C^2)^2 B^2 + D^2(D^2\lambda M + 2\lambda(\beta - 2A - C^2)\varpi)}}{D^2} \\ k_5 = \frac{2\lambda k_2 + k_3^2}{2\lambda\beta} \end{cases} \qquad (19)$$

For the particular solution, we can verify that $k_2 < 0$. To ensure $Q$ is concave, a property essential for verifying that this function indeed corresponds to the action-value function[1], we impose the additional conditions $k_0 < 0, k_0 k_2 - k_4^2 > 0$. Next, we state one of the main results of this paper.

**Theorem 3.** *Suppose the dynamics and the reward function are given by (16) and (17), respectively. Then, the Q-function is given by $Q(x, a) = \frac{1}{2}k_0 x^2 + k_1 x + \frac{1}{2}k_2 a^2 + k_3 a + k_4 xa + k_5$ where $k_0, k_1, k_2, k_3, k_4, k_5$ are as in (19). Furthermore, the optimal score function takes the form:*

$$\Psi^*(x, a) = \lambda^{-1}(k_2 a + k_3 + k_4 x). \qquad (20)$$

Additional details of Theorem 3 are provided in the Appendix C.

In our simulations, to ensure the stationarity of the controlled state process, we use the following model parameters: $A = -1, B = C = 0, D = 1, M = N = P' = 2, R = P = 1, \beta = 1, \lambda = 0.1$. We parameterize the Q-function as $Q^\theta = \frac{1}{2}\theta_0 x^2 + \theta_1 x + \theta_2 a^2 + \theta_3 a + \theta_4 xa + \theta_5$ and the corresponding score function as $\Psi^v(x, a) = -e^{v_1}a + v_2 x + v_3$. Using these parameterizations and the model setup, the optimal parameter values are computed as:

$$\theta^* = [-0.59047134, -0.23069812, -0.46141679, -0.35624157, -0.15119060, 0.17312350],$$
$$v^* = [1.52913155, -1.5119060, -3.5624157].$$

**Implementation Details.** The learning rate is initialized as $\alpha_\theta = \alpha_v = 0.01$ and decay according to $l(t) = \frac{1}{\max\{1, \sqrt{\log t}\}}$. To evaluate performance and stability, each experiment is repeated five times with different random seeds for sample generation. The parameter vector $\theta$ is initialized as zero and $v$ is initialized in the range $[0, 1]$. The corresponding optimal values $\theta^*$ and $v^*$ are then used as baselines for comparison. The time cost for each experiment is 352 seconds on a computer with Intel Core i5-10500 CPU and 32G Memory.

---

[1] the HJB equation has an additional quadratic solution, which, however, is convex.

**Performance Results.** Each experiment is repeated ten times with different random seeds. Figure 2 illustrates the convergence behavior of the proposed CQSM algorithm for one realized trajectory with time step $\Delta t = 0.1$. Both the Q-function and score function parameters gradually approach their theoretical optima except $\theta_2, \theta_3, v_2$. Note that these parameters are closely tied to $\nabla_a Q$. Our method requires the estimation of both the Q-function $Q$ and its gradient $\nabla_a Q$. This dual estimation introduces additional variance and bias, potentially leading to inaccurate policy updates.

Furthermore, we compare the performance of our CQSM algorithm against two benchmark methods in terms of the running average reward obtained during the learning process. The other two algorithms are the PG-based algorithm proposed in ([24], Algorithm 3) and the little q-learning algorithm presented in ([25], Algorithm 4). Figure 3 presents the running average rewards and standard deviations for all three methods under three step sizes, $\Delta t = 0.01, 0.1, 1$. Our proposed CQSM consistently outperforms the baselines in the early stages of training, achieving higher rewards more quickly than both PG and little q-learning. After a sufficient amount of time, all methods eventually stabilize to similar average reward levels.

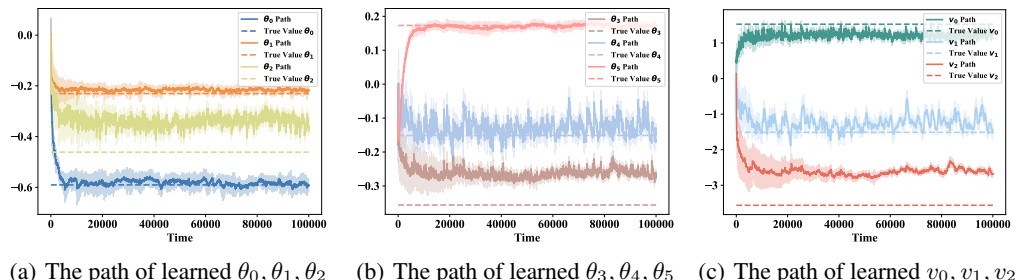

(a) The path of learned $\theta_0, \theta_1, \theta_2$ (b) The path of learned $\theta_3, \theta_4, \theta_5$ (c) The path of learned $v_0, v_1, v_2$

Figure 2: Paths of learned parameters of the CQSM reinforcement learning algorithm described in Algorithm 1. A single state trajectory of length $T = 10^5$ is generated. The dashed lines indicate the optimal parameter values. The shaded regions represent the standard deviation of the learned parameters across these runs, with the width of each shaded area equal to twice the corresponding standard deviation.

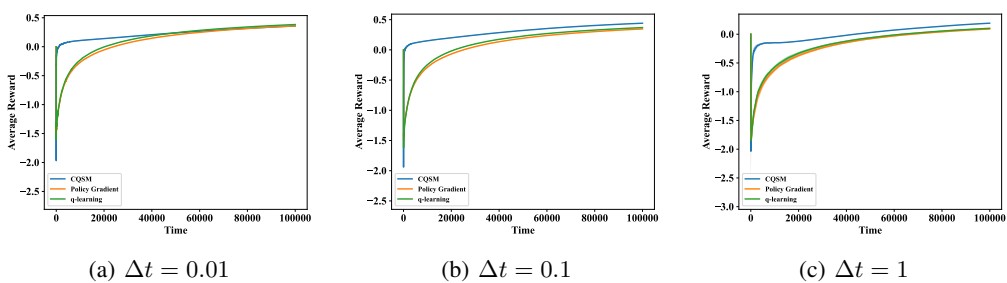

(a) $\Delta t = 0.01$  (b) $\Delta t = 0.1$  (c) $\Delta t = 1$

Figure 3: Running average rewards of three RL algorithms. A single state trajectory of length $T = 10^5$ is generated and discretized using three different step sizes: $\Delta t = 0.01$ in panel (a), $\Delta t = 0.1$ in panel (b), $\Delta t = 1$ in panel (c). For each setting, we apply three online algorithms: Policy Gradient described in Algorithm 3 in [24], q-Learning described in Algorithm 4 in [25] and CQSM described in Algorithm 1. We plot the mean running average reward over time and the shaded areas represent the standard deviation across the runs.

## 6 Conclusion

In this paper, we introduce a Q-function framework for continuous-time stochastic optimal control problems with diffusion policies. By using the dynamic programming principle, we derive the associated HJB equation for the Q-function. Building on this and utilizing the martingale orthogonality condition, we develop the CQSM algorithm. We further demonstrate the effectiveness of CQSM

in an LQ setting, showing promising results compared to existing continuous-time reinforcement learning algorithms.

Several interesting directions remain for future work. For the finite-horizon case, extending the approach to handle an important portfolio selection with a mean-variance objective poses a challenging problem due to inherent time inconsistency. Another promising direction is the optimization of the diffusion term $\sigma_a$, which could lead to improved exploration and performance. Furthermore, a theoretical convergence rate analysis of CQSM could offer deeper insights and guide further enhancements to the algorithm.

## Acknowledgements

We gratefully acknowledge financial support from the Key Project of National Natural Science Foundation of China 72432005, Guangdong Basic and Applied Basic Research Foundation 2023A1515030197.

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

# A  Diffusion RL: Problem Formulation and Well-Posedness

Listed below are the standard assumptions to ensure the well-posedness of the stochastic control problem in (1)-(3).

**Assumption 1.** *The following conditions for the dynamics and reward functions hold true:*
*(1) $b_X, \sigma_X, \sigma_a, r, h, \nabla_a r, \nabla_a h$ are all continuous functions in their respective arguments;*
*(2) $b_X, \sigma_X, \sigma_a$ are globally Lipschitz continuous in $(x, a)$, i.e., for $\varphi \in \{b, \sigma_X\}$, there exists a constant $C > 0$ such that*

$$\|\varphi(t, x, a) - \varphi(t, x', a')\|_2 \leq C(\|x - x'\|_2 + \|a - a'\|_2), \forall t \in [0, T], x, x' \in \mathbb{R}^n, a, a' \in \mathbb{R}^d;$$

*(3) $b_X, \sigma_X$ are linear growth continuous in $(x, a)$, i.e., for $\varphi \in \{b, \sigma_X\}$, there exists a constant $C > 0$ such that*

$$\|\varphi(t, x, a)\|_2 \leq C(\|x\|_2 + \|a\|_2), \forall t \in [0, T], x \in \mathbb{R}^n, a \in \mathbb{R}^d;$$

*(4) $\sigma_a$ is bounded for any $(x, a)$, i.e., there exist constants $C > 0$ such that $\|\sigma_a(t, x, a)\|_2 \leq C, \forall t \in [0, T], x \in \mathbb{R}^n, a \in \mathbb{R}^d$.*
*(5) $r, h, \nabla_a r, \nabla_a h$ have polynomial growth in $(x, a)$, i.e., for $\varphi \in \{r, h, \nabla_a r, \nabla_a h\}$ there exist constants $C > 0$ such that*

$$|\varphi(t, x, a)| \leq C(1 + \|x\|_2 + \|a\|_2), \forall t \in [0, T], x \in \mathbb{R}^n, a \in \mathbb{R}^d.$$

**Lemma 1.** *Suppose $\{a_t : t \geq 0\}$ follows*

$$\mathrm{d}a_t = b_t \, \mathrm{d}t + \sigma_t \, \mathrm{d}B_t^a, t \geq 0,$$

*with $C_\sigma := \mathrm{ess\,sup}_{t,\omega} |\sigma_t| < \infty$. If $\int_0^t |b_s|^2 \, \mathrm{d}s < \infty$ for all $t \geq 0, a \in \mathbb{R}^d$, then there exists a constant $C > 0$, which is independent of $T$ and $a_0$, such that*

$$\mathbb{E}^{\mathbb{P}} \left[ \sup_{0 \leq t \leq T} |a_t|^2 \right] \leq C \left( 1 + |a_0|^2 \right), \forall T \geq 0. \tag{21}$$

*Proof.* By the elementary inequality

$$(a + b + c)^2 \leq (3 \max\{a, b, c\})^2 \leq 3^2 a^2 + 3^2 b^2 + 3^2 c^2, a, b, c \geq 0,$$

we have

$$\mathbb{E}^{\mathbb{P}} \left[ \sup_{0 \leq t \leq T} |a_t|^2 \right] \leq \mathbb{E}^{\mathbb{P}} \left[ \left( |a_0| + \sup_{0 \leq t \leq T} \int_0^t |b_s| \, \mathrm{d}s + \sup_{0 \leq t \leq T} \left| \int_0^t \sigma_s \, \mathrm{d}B_s^a \right| \right)^2 \right]$$

$$\leq \mathbb{E}^{\mathbb{P}} \left[ \left( |a_0| + \int_0^T |b_s| \, \mathrm{d}s + \sup_{0 \leq t \leq T} \left| \int_0^t \sigma_s \, \mathrm{d}B_s^a \right| \right)^2 \right]$$

$$\leq 3^2 |a_0|^2 + 3^2 \int_0^T |b_s|^2 \, \mathrm{d}s + 3^2 \mathbb{E}^{\mathbb{P}} \left[ \sup_{0 \leq t \leq T} \left| \int_0^t \sigma_s \, \mathrm{d}B_s^a \right|^2 \right]$$

$$\leq 3^2 |a_0|^2 + 3^2 \int_0^T |b_s|^2 \, \mathrm{d}s + 3^2 C_2 \left( \mathbb{E}^{\mathbb{P}} \left[ \int_0^t |\sigma_s|^2 \, \mathrm{d}s \right] \right)$$

$$\leq 3^2 |a_0|^2 + 3^2 \int_0^T |b_s|^2 \, \mathrm{d}s + 3^2 C_2 C_\sigma^2 T$$

$$\leq C \left( 1 + |a_0|^2 \right),$$

where the second to last inequality is due to the Burkholder–Davis–Gundy inequality. This proves (21). $\qquad \square$

**Lemma 2.** *Let Assumption 1 hold, the solution of state SDE (1) satisfies the condition*

$$\mathbb{E}^{\mathbb{P}} \left[ \sup_{0 \leq t \leq T} |X_t|^2 \right] \leq C \left( 1 + |x_0|^2 \right), \tag{22}$$

*for some constant $C > 0$.*

*Proof.* Based on the proved growth condition on $b_X, \sigma_X$, Cauchy–Schwarz inequality, and Burkholder-Davis-Gundy inequalities, we obtain

$$\mathbb{E}^{\mathbb{P}}\left[\sup_{0 \leq t \leq T} |X_t|^2\right]$$

$$\leq C_1 \mathbb{E}^{\mathbb{P}}\left[|x_0|^2 + \sup_{0 \leq t \leq T}\left|\int_0^t b_X(s, X_s, a_s)\,\mathrm{d}s\right|^2 + \sup_{0 \leq t \leq T}\left|\int_0^t \sigma_X(s, X_s, a_s)\,\mathrm{d}B_s^X\right|^2\right]$$

$$\leq C_1 \mathbb{E}^{\mathbb{P}}\left[|x_0|^2 + C_2 \int_0^T \left(\sup_{0 \leq \tau \leq s} |X_\tau|^2 + |a_s|^2\right)\mathrm{d}s\right]$$

$$\leq C_3(1 + |x_0|^2) + C_4 \int_0^T \sup_{0 \leq \tau \leq s} \mathbb{E}^{\mathbb{P}}\left[|X_\tau|^2\right]\mathrm{d}s.$$

Applying Gronwall's inequality, we obtain the desired result. $\qquad\square$

**Theorem 4.** *Let Assumption 1 hold, then there exists a constant $C_1 > 0$ such that the Q-function satisfies*

$$|Q(t, x, a)| \leq C_1(1 + \|x\|_2 + \|a\|_2),$$

*for all $t \in [0, T], x \in \mathbb{R}^n, a \in \mathbb{R}^d$. Finally, the Q-function is finite.*

*Proof.* Let $\Psi = C$ where $C$ is a constant. It then follows from Lemma 1 and Lemma 2 that

$$Q(t, x, a)$$

$$\geq \mathbb{E}^{\mathbb{P}}\left[\int_t^T \left[r(s, X_s, a_s) - \frac{\lambda}{2}\|\Psi(s, X_s, a_s)\|_2^2\right]\mathrm{d}s + h(X_T, a_T)\Big| X_t = x, a_t = a\right]$$

$$\geq \mathbb{E}\left[\int_t^T \left(-C(1 + \|X_s\|_2 + \|a_s\|_2) - \frac{\lambda}{2}C^2\right)\mathrm{d}s\Big| X_t = x, a_t = a\right]$$

$$\geq -C'(1 + \|x\|_2 + \|a\|_2)$$

for some constant $C'$ independent of $x, a$. On the other hand, for any $\Psi \in \Pi$, we have

$$Q(t, x, a) = \mathbb{E}^{\mathbb{P}}\left[\int_t^T \left[r(X_s, a_s^p) - \frac{\lambda}{2}\|\Psi(X_s, a_s^p)\|_2^2\right]\mathrm{d}s + h(X_T, a_T)\Big| X_t = x, a_t = a\right]$$

$$\leq \mathbb{E}^{\mathbb{P}}\left[\int_t^T [C(1 + \|X_s\|_2 + \|a_s\|_2)]\,\mathrm{d}s\Big| X_t = x, a_t = a\right]$$

$$\leq C''(1 + \|x\|_2 + \|a\|_2)$$

for some constant $C''$ independent of $x, a$. The final result is evident. The proof is complete. $\qquad\square$

We have indeed established in the above that

$$\mathbb{E}^{\mathbb{P}}\left[\int_t^T |r(s, X_s, a_s)|\,\mathrm{d}s\right] < \infty. \tag{23}$$

# B  Proofs of Martingale Characterization and Score Improvement Theorem

## B.1  Proof of Theorem 1

To show $M_s = Q(s, X_s, a_s; \Psi) + \int_t^s \left[r(u, X_u, a_u) - \frac{1}{2}\lambda\|\Psi(u, X_u, a_u)\|_2^2\right]\mathrm{d}u$ is a martingale, observe that

$$M_s = \mathbb{E}^{\mathbb{P}}\left[\int_s^T \left[r(u, X_u, a_u) - \frac{1}{2}\lambda\|\Psi(u, X_u, a_u)\|_2^2\right]\mathrm{d}s + h(X_T, a_T)\Big| X_s, a_s\right]$$

$$+ \int_t^s \left[r(u, X_u, a_u) - \frac{1}{2}\lambda\|\Psi(u, X_u, a_u)\|_2^2\right]\mathrm{d}u \tag{24}$$

$$= \mathbb{E}^{\mathbb{P}}\left[M_T | \mathcal{F}_s\right],$$

where we have used the Markov property of the process $\{(X_s, a_s), t \le s \le T\}$. This establishes that $M$ is a martingale.

Conversely, if $\tilde{M}$ is a martingale, then $\tilde{M}_s = \mathbb{E}^{\mathbb{P}}\left[\tilde{M}_T | \mathcal{F}_s\right]$, which is equivalent to

$$
\begin{aligned}
\tilde{Q}(s, X_s, a_s; \Psi) &= \mathbb{E}^{\mathbb{P}}\left[\int_s^T \left[r(u, X_u, a_u) - \frac{1}{2}\lambda \|\Psi(u, X_u, a_u)\|_2^2\right] \mathrm{d}u + \tilde{Q}(T, X_T, a_T)\Big|\mathcal{F}_s\right] \\
&= \mathbb{E}^{\mathbb{P}}\left[\int_s^T \left[r(u, X_u, a_u) - \frac{1}{2}\lambda \|\Psi(u, X_u, a_u)\|_2^2\right] \mathrm{d}u + h(X_T)\Big|\mathcal{F}_s\right] \\
&= Q(s, X_s, a_s; \Psi), s \in [t, T].
\end{aligned}
\tag{25}
$$

Letting $s = t$, we conclude $\tilde{Q}(t, x, a; \Psi) = Q(t, x, a; \Psi)$.

The "only if" part is evident. To prove the "if" part, assume that $\mathrm{d}M_t = A_t\,\mathrm{d}t + C_t\,\mathrm{d}B_t$. In particular, in our case, $A_t = \mathcal{L}Q(t, x, a; \Psi) + r(t, X_t, a_t) - \frac{1}{2}\lambda \|\Psi(t, X_t, a_t)\|_2^2$ and $C_t = \left(\frac{\partial Q}{\partial Z}\right)^\top G(t, Z_t)$. $A, C \in L^2_{\mathcal{F}}([0, T])$ follows by assumption ($Q \in C^{1,2,2}([0, T) \times \mathbb{R}^n \times \mathbb{R}^d) \cap C([0, T) \times \mathbb{R}^n \times \mathbb{R}^d)$) and Theorem 4. For any $0 \le s < s' \le T$, take $\xi_t = sgn(A_t)$ if $t \in [s, s']$ and $\xi_t = 0$ otherwise. Then

$$
0 = \mathbb{E}^{\mathbb{P}}\int_s^{s'} \xi_t\,\mathrm{d}M_t = \mathbb{E}^{\mathbb{P}}\int_s^{s'} (|A_t|\,\mathrm{d}t + \xi_t C_t\,\mathrm{d}B_t) = \mathbb{E}^{\mathbb{P}}\int_s^{s'} |A_t|\,\mathrm{d}t,
\tag{26}
$$

where the expectation of the second term vanishes because $|\xi C| \le |C| \in L^2_{\mathcal{F}}([0, T])$ and hence $\mathbb{E}^{\mathbb{P}}\int_0^\cdot \xi_t C_t\,\mathrm{d}B_t$ is a martingale. This yields $A_t = 0$ almost surely, and thus $M$ is a martingale.

## B.2 Proof of Theorem 2

Fix $(x, a) \in \mathbb{R}^n \times \mathbb{R}^d$, applying Ito's formula, we have

$$
\begin{aligned}
e^{-\beta s}Q(X_s, a_s; \Psi) =\ & e^{-\beta t}Q(x, a; \Psi) + \int_t^s e^{-\beta(\tau-t)}\Big\{-\beta Q(X_\tau, a_\tau; \Psi) + \nabla_x Q^\top \cdot b_X(X_\tau, a_\tau) \\
& + \nabla_a Q^\top \cdot \Psi(X_\tau, a_\tau)) + \frac{1}{2}\mathrm{tr}\left(\sigma_X \sigma_X^\top \nabla_x^2 Q\right) + \frac{1}{2}\mathrm{tr}\left(\sigma_a \sigma_a^\top \nabla_a^2 Q\right)\Big\}\mathrm{d}\tau \\
& + \int_t^s e^{-\beta(\tau-t)}\nabla_x Q^\top \cdot \sigma_X(X_\tau, a_\tau)\,\mathrm{d}B_\tau^X + e^{-\beta\tau}\nabla_a Q^\top \cdot \sigma_a(X_\tau, a_\tau)\,\mathrm{d}B_\tau^a.
\end{aligned}
\tag{27}
$$

Define the stopping times $T_n := \inf\{s \ge t : \|X_s\|_2 \ge n, \|a_s\|_2 \ge n\}$, for $n \ge 1$. Then we have

$$
\begin{aligned}
\mathbb{E}^{\mathbb{P}}\Big[e^{-\beta(s \wedge T_n)}&Q(X_{s \wedge T_n}, a_{s \wedge T_n}; \Psi)\Big|X_t = x, a_t = a\Big] = e^{-\beta t}Q(x, a; \Psi) \\
& + \mathbb{E}^{\mathbb{P}}\bigg[\int_t^{s \wedge T_n} e^{-\beta(\tau-t)}\Big\{-\beta Q(X_\tau, a_\tau; \Psi) + \nabla_x Q^{\Psi,\top} \cdot b_X(X_\tau, a_\tau) + \nabla_a Q^\top \cdot \Psi(X_\tau, a_\tau) \\
& + \frac{1}{2}\mathrm{tr}\left(\sigma_X \sigma_X^\top \nabla_x^2 Q\right) + \frac{1}{2}\mathrm{tr}\left(\sigma_a \sigma_a^\top \nabla_a^2 Q\right)\Big\}\mathrm{d}\tau\Big|X_t = x, a_t = a\bigg].
\end{aligned}
\tag{28}
$$

On the other hand, by standard arguments and the assumption that $Q(\cdot, \cdot; \Psi)$ is smooth, we have

$$
\begin{aligned}
\beta Q(x, a; \Psi) - &\left\{\nabla_x Q^\top \cdot b_X(x, a) + \nabla_a Q^\top \cdot \Psi(x, a)\right. \\
& \left.+ \frac{1}{2}\mathrm{tr}\left(\sigma_X \sigma_X^\top \nabla_x^2 Q\right) + \frac{1}{2}\mathrm{tr}\left(\sigma_a \sigma_a^\top \nabla_a^2 Q\right) + r(x, a) - \frac{1}{2}\lambda\|\Psi\|_2^2\right\} = 0,
\end{aligned}
\tag{29}
$$

for any $(x, a) \in \mathbb{R}^n \times \mathbb{R}^d$. It follows that

$$
\begin{aligned}
\beta Q(x, a; \Psi) - \sup_{\tilde{\Psi}}&\left\{\nabla_x Q^{\Psi,\top} \cdot b_X(x, a) + \nabla_a Q^\top \cdot \tilde{\Psi}(x, a)\right. \\
& \left.+ \frac{1}{2}\mathrm{tr}\left(\sigma_X \sigma_X^\top \nabla_x^2 Q\right) + \frac{1}{2}\mathrm{tr}\left(\sigma_a \sigma_a^\top \nabla_a^2 Q\right) + r(x, a) - \frac{1}{2}\lambda\|\tilde{\Psi}\|_2^2\right\} \le 0.
\end{aligned}
\tag{30}
$$

Notice that the minimizer of the Hamiltonian in (30) is given by $\Psi^1 = \lambda^{-1}\nabla_a Q(x, a; \Psi)$ for some $\lambda > 0$. It then follows that Equation (28) implies

$$\mathbb{E}^{\mathbb{P}}\left[e^{-\beta(s\wedge T_n)}Q(X_{s\wedge T_n}, a_{s\wedge T_n}; \Psi)\Big|X_t = x, a_t = a\right]$$

$$\geq e^{-\beta t}Q(x, a; \Psi) - \mathbb{E}^{\mathbb{P}}\left[\int_t^{s\wedge T_n} e^{-\beta(\tau-t)}\left(r(x_\tau, a_\tau) - \frac{\lambda}{2}\|\Psi^1\|_2^2\right)\mathrm{d}\tau\Big|X_t = x, a_t = a\right]. \tag{31}$$

Sending $n \to \infty$, we deduce that

$$\mathbb{E}^{\mathbb{P}}\left[e^{-\beta s}Q(X_s, a_s; \Psi)\big|X_t = x, a_t = a\right]$$

$$\geq e^{-\beta t}Q(x, a; \Psi) - \mathbb{E}^{\mathbb{P}}\left[\int_t^s e^{-\beta(\tau-t)}\left(r(x_\tau, a_\tau) - \frac{\lambda}{2}\|\Psi^1\|_2^2\right)\mathrm{d}\tau\Big|X_t = x, a_t = a\right]. \tag{32}$$

Noting Lemma 1, Lemma 2 and $Q$ is polynomial growth, we have

$$\liminf_{s\to\infty}\mathbb{E}^{\mathbb{P}}\left[e^{-\beta s}Q(X_s, a_s; \Psi)\big|X_t = x, a_t = a\right]$$

$$\leq \limsup_{s\to\infty}\mathbb{E}^{\mathbb{P}}\left[e^{-\beta s}Q(X_s, a_s; \Psi)\big|X_t = x, a_t = a\right] = 0 \tag{33}$$

and applying the dominated convergence theorem yield

$$Q(x, a; \Psi) \leq \mathbb{E}^{\mathbb{P}}\left[\int_t^\infty e^{-\beta\tau}\left(r(x_\tau, a_\tau) - \frac{\lambda}{2}\|\Psi^1\|_2^2\right)\mathrm{d}\tau\Big|X_t = x, a_t = a\right] = Q(x, a; \Psi^1). \tag{34}$$

## C    Derivation of Linear-Quadratic Stochastic Control

The following derivation corresponds to Section 5 of the main text.

**Assumption 2.** *The discounted rate satisfies $\beta > 2A + C^2$.*

This assumption ensures a sufficiently large discount rate, which guarantees that $\liminf_{T\to\infty} e^{-\beta T}\mathbb{E}^{\mathbb{P}}[Q(X_T, a_T; \Psi)] = 0$ for any score $\Psi$, thereby ensuring the corresponding expected reward remains finite.

By HJB equation

$$\beta Q(x, a) - Q_x b(x, a) - \frac{1}{2\lambda}Q_a^2 - \frac{1}{2}\sigma_X^2 Q_{xx} - \frac{1}{2}\sigma_a^2 Q_{aa} - r(x, a) = 0, \tag{35}$$

we have

$$x^2: \quad \frac{\beta}{2}k_0 - Ak_0 - \frac{1}{2\lambda}k_4^2 - \frac{1}{2}C^2 k_0 + M/2 = 0,$$

$$x: \quad \beta k_1 - Ak_1 - \frac{k_3 k_4}{\lambda} + P = 0,$$

$$a^2: \quad \frac{\beta}{2}k_2 - k_4 B - \frac{1}{2\lambda}k_2^2 - \frac{1}{2}k_0 D^2 + N/2 = 0,$$

$$a: \quad \beta k_3 - Bk_1 - \frac{k_2 k_3}{\lambda} + P' = 0,$$

$$xa: \quad \beta k_4 - k_0 B - k_4 A - \frac{1}{\lambda}k_2 k_4 - k_0 CD + R = 0,$$

$$\text{Cons}: \quad \beta k_5 - k_2 - \frac{1}{2\lambda}k_3^2 = 0.$$

By $x^2$ term:

$$k_0 = \frac{1}{\lambda(\beta - 2A - C^2)}k_4^2 - \frac{M}{\beta - 2A - C^2} \tag{36}$$

and substitute $k_0$ to $a^2$ term, we obtain

$$\frac{\beta}{2}k_2 - k_4 B - \frac{1}{2\lambda}k_2^2 - \frac{1}{2}D^2\left(\frac{1}{\lambda(\beta - 2A - C^2)}k_4^2 - \frac{M}{\beta - 2A - C^2}\right) + N/2 = 0, \tag{37}$$

$$\frac{\beta}{2}k_2 - \frac{1}{2\lambda}k_2^2 + N/2 = k_4 B + \frac{1}{2}D^2\left(\frac{1}{\lambda(\beta - 2A - C^2)}k_4^2 - \frac{M}{\beta - 2A - C^2}\right) := \varpi. \tag{38}$$

First, we consider $D \neq 0$. Hence, we have

$$k_2 = \frac{\beta}{2}\lambda - \lambda\sqrt{\frac{\beta^2}{4} + \frac{1}{\lambda}(N - 2\varpi)}, \tag{39}$$

$$k_4 = \frac{-\lambda(\beta - 2A - C^2)B \pm \sqrt{\lambda^2(\beta - 2A - C^2)^2 B^2 + D^2(D^2\lambda M + 2\lambda(\beta - 2A - C^2)\varpi)}}{D^2}. \tag{40}$$

By $xa$ term, we can determine the value of $\varpi$ and $\varpi$ satisfies the following bounds

$$\frac{\beta^2}{4} + \frac{1}{\lambda}(N - 2\varpi) \geq 0, \tag{41}$$

$$\lambda^2(\beta - 2A - C^2)^2 B^2 + D^2(D^2\lambda M + 2\lambda(\beta - 2A - C^2)\varpi) \geq 0. \tag{42}$$

If $D = 0$, then

$$k_2 = \frac{\beta\lambda}{2} - \lambda\sqrt{\frac{\beta^2}{4} - \frac{2}{\lambda}\left(\frac{N}{2} - k_4 B\right)}, \tag{43}$$

and $k_4$ satisfies the following equation:

$$-\frac{B}{\lambda(\beta - 2A)}k_4^2 + \left(\frac{\beta}{2} - A + \sqrt{\frac{\beta^2}{4} - \frac{2}{\lambda}\left(\frac{N}{2} - k_4 B\right)}\right) k_4 + \frac{MB}{\beta - 2A} + R = 0. \tag{44}$$

Furthermore, by $x$ term and $a$ term, we have

$$k_1 = \frac{1}{\lambda(\beta - A)}k_3 k_4 - \frac{P}{\beta - A}, \tag{45}$$

$$k_3 = -\frac{BP + P'(\beta - A)}{(\beta - A)\left(\beta - \frac{B}{\lambda(\beta - A)}k_4 - \frac{1}{\lambda}k_2\right)}. \tag{46}$$

Finally, we have

$$k_5 = \frac{2\lambda k_2 + k_3^2}{2\lambda\beta}. \tag{47}$$

Therefore, the optimal score function takes the form:

$$\Psi^*(x, a) = \lambda^{-1}\nabla_a Q(x, a) = \lambda^{-1}(k_2 a + k_3 + k_4 x). \tag{48}$$

# D  Continuous Actor-Critic Q-Learning Algorithms

**Score Evaluation of the Q-Function.**  We provide a complete description of how the HJB equation is used to construct the continuous-time Q-learning algorithm. For estimating $Q(x, a; \Psi)$, a number of algorithms can be developed based on two types of objectives: to minimize the martingale loss function or to satisfy the martingale orthogonality conditions. We summarize these methods in the Q-learning context below.

(1) Minimize the martingale loss function:

$$\frac{1}{2}\mathbb{E}^{\mathbb{P}}\left[\int_0^T \left[h(X_T, a_T) - Q^\theta(t, X_t, a_t) + \int_t^T \left(r(s, X_s, a_s) - \frac{\lambda}{2}\|\Psi^v(s, X_s, a_s)\|^2\right) ds\right]^2 dt\right]. \tag{49}$$

This method is intrinsically offline because the loss function involves the whole horizon $[0, T]$. We can apply stochastic gradient decent to update

$$\theta \leftarrow \theta + \alpha_\theta \int_0^T \frac{\partial Q^\theta}{\partial \theta}(t, X_t, a_t)G_{t:T}\, dt,$$

$$v \leftarrow v + \alpha_v \int_0^T \int_t^T \lambda\Psi^v(s, X_s, a_s)\frac{\partial\Psi^v}{\partial v}(s, X_s, a_s)\, ds G_{t:T}\, dt, \tag{50}$$

where $G_{t:T} = h(X_T, a_T) - Q^\theta(t, X_t, a_t) + \int_t^T \left( r(s, X_s, a_s) - \frac{\lambda}{2} \|\Psi^v(s, X_s, a_s)\|^2 \right) \mathrm{d}s$. We present Algorithm 2 based on this updating rule. Note that this algorithm is analogous to the classical gradient Monte Carlo method or TD(1) for MDPs [44] because full sample trajectories are used to compute gradients.

(2) We leverage the martingale orthogonality condition, which states that for any $T > 0$ and a suitable test process $\xi$,

$$\mathbb{E}^{\mathbb{P}} \int_0^T \xi_t \left\{ \mathrm{d}Q^\theta(t, X_t, a_t; \Psi) + \left[ r(t, X_t, a_t) - \frac{1}{2}\lambda\|\Psi^v(t, X_t, a_t)\|_2^2 \right] \mathrm{d}t \right\} = 0. \qquad (51)$$

We use stochastic approximation to update $\theta$ either offline by

$$\theta \leftarrow \theta + \alpha_\theta \int_0^T \xi_t \left\{ \mathrm{d}Q^\theta(t, X_t, a_t; \Psi) + \left[ r(t, X_t, a_t) - \frac{1}{2}\lambda\|\Psi^v(t, X_t, a_t)\|_2^2 \right] \mathrm{d}t \right\}, \qquad (52)$$

or online by

$$\theta \leftarrow \theta + \alpha_\theta \xi_t \left\{ \mathrm{d}Q^\theta(t, X_t, a_t; \Psi) + \left[ r(t, X_t, a_t) - \frac{1}{2}\lambda\|\Psi^v(t, X_t, a_t)\|_2^2 \right] \mathrm{d}t \right\}. \qquad (53)$$

Typical choices of test functions are $\xi_t = \frac{\partial Q^\theta}{\partial \theta}$ or $\xi_t = \int_0^t \rho^{s-t} \frac{\partial Q^\theta}{\partial \theta} \, \mathrm{d}s, 0 < \rho \leq 1$ which lead to Q-learning algorithms based on stochastic approximation. However, relying solely on this specific choice does not, in general, guarantee satisfaction of the full martingale condition. Moreover, the convergence of the resulting stochastic approximation algorithm is not assured without additional assumptions. As discussed in [23], the selection of test functions must be carefully tailored to the structure of the Q-function, highlighting the need for more robust and theoretically grounded choices in continuous-time settings.

(3) Choose the same type of test functions $\xi_t$ as above but now minimize the GMM objective functions:

$$\mathbb{E}^{\mathbb{P}} \left[ \int_0^T \xi_t \left[ \mathrm{d}J^\theta(t, X_t^{\pi^\psi}) + r(t, X_t^{\pi^\psi}, a_t^{\pi^\psi}) \mathrm{d}t - q^\psi(t, X_t^{\pi^\psi}, a_t^{\pi^\psi}) \mathrm{d}t - \beta J^\theta(t, X_t^{\pi^\psi}) \mathrm{d}t \right]^\top \right]$$

$$A_\theta \mathbb{E}^{\mathbb{P}} \left[ \int_0^T \xi_t \left[ \mathrm{d}J^\theta(t, X_t^{\pi^\psi}) + r(t, X_t^{\pi^\psi}, a_t^{\pi^\psi}) \mathrm{d}t - q^\psi(t, X_t^{\pi^\psi}, a_t^{\pi^\psi}) \mathrm{d}t - \beta J^\theta(t, X_t^{\pi^\psi}) \mathrm{d}t \right] \right],$$

$$(54)$$

where $A_\theta \in \mathbb{S}^{L_\theta}$. Typical choices of these matrices are $A_\theta = I_{L_\theta}$ or $A_\theta = (\mathbb{E}^{\mathbb{P}}[\int_0^T \xi_t \xi_t^\top \mathrm{d}t])^{-1}$. Again, we refer the reader to [23] for discussions on these choices and the connection with the classical GTD algorithms and GMM method.

**Score Gradient.** We aim to compute the score gradient $g(x, a; v) := \frac{\partial}{\partial v}Q(x, a; \Psi^v) \in \mathbb{R}^{L_v}$ at the current state-action pair $(x, a)$. Based on the HJB of the Q-function, we take the derivative in $v$ on both sides to get

$$\mathcal{L}g(x, a; v) - \beta g(x, a; v) + (\nabla_a Q(x, a; \Psi^v) - \lambda\Psi^v(x, a))\frac{\partial \Psi^v}{\partial v}(x, a) = 0. \qquad (55)$$

Thus, a Feynman-Kac formula represents $g$ as

$$g(x, a; v) = \mathbb{E}^{\mathbb{P}} \left[ \int_t^\infty e^{-\beta s}(\nabla_a Q(X_s, a_s; \Psi^v) - \lambda\Psi^v(X_s, a_s))\frac{\partial \Psi^v}{\partial v}(X_s, a_s) \, \mathrm{d}s \bigg| X_t = x, a_t = a \right]. \qquad (56)$$

To treat the online case, assume that $v^*$ is the optimal point of $Q(x, a; \Psi^v)$ for any $(x, a)$ and that the first-order condition holds (e.g., when $v^*$ is an interior point). Then $g(x, a; v^*) = 0$. It follows that

$$0 = \mathbb{E}^{\mathbb{P}} \left[ \int_t^\infty \eta_s(\nabla_a Q(X_s, a_s; \Psi^v) - \lambda\Psi^v(X_s, a_s))\frac{\partial \Psi^v}{\partial v}(X_s, a_s) \, \mathrm{d}s \bigg| X_t = x, a_t = a \right], \qquad (57)$$

for any $\eta \in L^2_{\mathcal{F}}([0, T]; Q(\cdot, X., a.; \Psi))$. If we take $\eta_s = e^{-\beta s}$, then the right hand side (57) coincides with $g(x, a; v^*)$. More importantly, besides the flexibility of choosing different sets of test functions,

(57) provides a way to derive a system of equations based on only past observations and, hence, enables online learning. For example, by taking $\eta_s = 0$ on $[T, \infty]$, (57) involves sample trajectories up to time $T$. Thus, learning the optimal policy either offline or online boils down to solving a system of equations (with suitably chosen test functions) via stochastic approximation to find $v^*$. Online learning of (57) is the same as the update rule: $v \in \arg\min \frac{1}{2} \|\Psi^v(x, a) - \lambda^{-1} \nabla_a Q(x, a)\|^2$.

Here we present Offline Continuous Q-Score Matching (CQSM) Martingale Loss Algorithm 2 in the infinite-horizon setting .

---

**Algorithm 2** Offline Continuous Q-Score Matching (CQSM) Martingale Loss Algorithm

---

1: **Inputs:** Initial state $x_0, a_0$, horizon $T$, time step $\Delta t$, number of episodes $N$, number of mesh grids $K$, initial learning rates $\alpha_\theta, \alpha_v$ and a learning rate schedule function $l(\cdot)$, functional forms of parameterized action value function $Q^\theta(\cdot, \cdot)$ and score function $\Psi^v(\cdot, \cdot)$ and regularization parameter $\lambda$.
2: Initialize $\theta, v$.
3: **for** episode $j = 1$ to $N$ **do**
4:     Initialize $k = 0$. Observe initial state $x_0$ and store $x_{t_k} \leftarrow x_0$.
5:     Choose action by iteratively denoising $a^T \sim \mathcal{N}(0, 1) \rightarrow a^0$ using $\Psi^v : a_{t_k} \sim \pi^v(x_{t_k})$;
6:     Step environment $\{r_{t_k}, x_{t_{k+1}}\} = env(a_{t_k})$.
7:     Obtain one observation $\{a_{t_k}, r_{t_k}, x_{t_k}\}_{k=0,\cdots,K-1}$.
8:     For every $k = 0$ to $K - 1$, compute

$$G_{t_k:T} = -e^{-\beta t_k} Q^\theta(x_{t_k}, a_{t_k}) + \sum_{i=k}^{K-1} e^{-\beta t_i} [r(x_{t_i}, a_{t_i}) - \frac{\lambda}{2} \|\Psi^v(x_{t_i}, a_{t_i})\|^2] \Delta t. \tag{58}$$

9:     Update $\theta$ and $v$ by

$$\theta \leftarrow \theta + l(j)\alpha_\theta \sum_{k=0}^{K-1} \frac{\partial Q^\theta}{\partial \theta}(x_{t_k}, a_{t_k}) G_{t_k:T} \Delta t. \tag{59}$$

$$v \leftarrow v + l(j)\alpha_v \sum_{k=0}^{K-1} \left[ \sum_{i=k}^{K-1} \lambda \Psi^v(x_{t_i}, a_{t_i}) \frac{\partial \Psi^v}{\partial v}(x_{t_i}, a_{t_i}) \Delta t \right] G_{t_k:T} \Delta t. \tag{60}$$

10: **end for**

---

# E  Scored Based Diffusion Models

Consider the forward SDE with state space $Y_t \in \mathbb{R}^d$ is defined as

$$\mathrm{d}Y_t = f(t, Y_t)\,\mathrm{d}t + g(t)\,\mathrm{d}B_t^a, Y_0 \sim \pi_{\text{data}}(\cdot), \tag{61}$$

where $f : \mathbb{R}_+ \times \mathbb{R}^d \rightarrow \mathbb{R}^d$ and $g : \mathbb{R}_+ \rightarrow \mathbb{R}_+$. Denote $\pi(t, \cdot)$ as the probability density of $Y_t$.

Set time horizon $T > 0$ to be fixed, and run the SDE (61) until time $T$ to get $Y_T \sim \pi(T, \cdot)$. The time reversal $Y_t^{rev} := Y_{T-t}$ for $0 \leq t \leq T$ satisfies an SDE, under some mild conditions on $f$ and $g$:

$$\mathrm{d}Y_t^{rev} = \left( -f(T - t, Y^{rev}) + \frac{1 + \eta^2}{2} g^2(T - t) \nabla_y \log \pi(T - t, Y^{rev}) \right) \mathrm{d}t + \eta g(T - t)\,\mathrm{d}B_t^a, \tag{62}$$

where $\nabla_y \log \pi(t, y)$ is known as the stein score function and $\eta \in [0, 1]$ is a constant. In addition, a special but important case by taking $\eta = 0$ in the (62), this results to a flow ODE [42]:

$$\mathrm{d}Y_t^{rev} = \left( -f(T - t, Y^{rev}) + \frac{1}{2} g^2(T - t) \nabla_y \log \pi(T - t, Y^{rev}) \right) \mathrm{d}t, \tag{63}$$

which can enable faster sampling and likelihood computation thanks to the deterministic generation.

Since the score function $\nabla_y \log \pi(t, y)$ is unknown, diffusion models learn a function approximation $s_\theta(t, y)$, parameterized by $\theta$ (usually a neural network), to the true score by minimizing the MSE or

the Fisher divergence (between the learned distribution and true distribution), evaluated by samples generated through the forward process (61):

$$\theta^* = \arg\max_{\theta} \mathbb{E}_{t\sim\text{Uni}(0,T)} \left\{ \lambda(t)\mathbb{E}_{y_0\sim\pi_{\text{data}}}\mathbb{E}_{y_t\sim\pi(t,\cdot|y_0)} \left[ \|s_\theta(t,y_t) - \nabla_{y_t}\log\pi(t,y_t|y_0)\|_2^2 \right] \right\} \quad (64)$$

where $\lambda : [0,T] \to \mathbb{R}_{>0}$ is a chosen positive weighting function. When choosing the weighting functions as $\lambda(t) = g^2(t)$, this score matching objective is equivalent to maximizing an evidence lower bound (ELBO) of the log-likelihood.

For action sampling, we view the action process as the reverse process. First we set $\sigma_a(t, X, a^t) = \sigma_t^2 I$ to untrained time dependent constants. Experimentally, we set $\beta_t = \sigma_t^2$ and $\alpha_t = 1 - \beta_t, \bar{\alpha}_t = \prod_{s=1}^t \alpha_s$. When applied to score matching with denoising diffusion probabilistic modeling (DDPM) [18], samples can be generated by starting from $a^T \sim \mathcal{N}(0, I)$ and following the reverse Markov chain as below

$$a^{t-1} = \frac{1}{\sqrt{\alpha_t}}\left( a^t + \frac{1 - \alpha_t}{\sqrt{1 - \bar{\alpha}_t}}\Psi(a_t, x_t) \right) + \sigma_t Z, Z \sim \mathcal{N}(0, I). \quad (65)$$

## F   Additional Experiments

### F.1   Deterministic Case of LQ Control Tasks

We set $C = D = 0$ to eliminate stochasticity. Figure F.1 shows the running average reward of CQSM, PG and little q-learning. We observe that the resulting performance is comparable to the stochastic case.

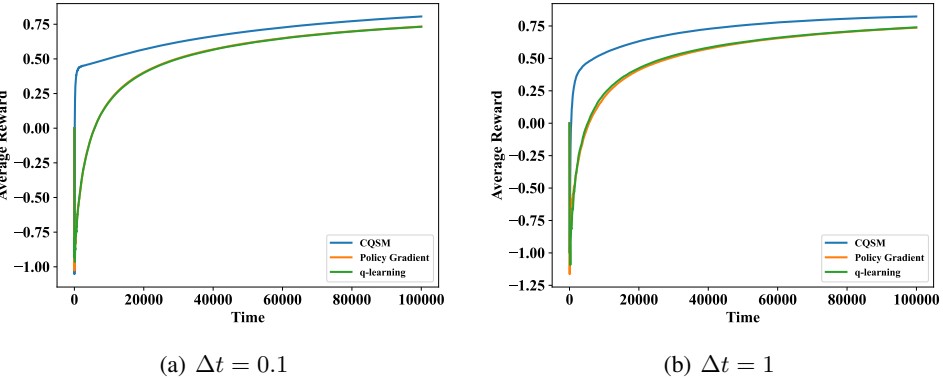

(a) $\Delta t = 0.1$               (b) $\Delta t = 1$

Figure F.1: Running average rewards of three RL algorithms. A single state trajectory of length $T = 10^5$ is generated and discretized using two different step sizes: $\Delta t = 0.1$ in panel (a), $\Delta t = 1$ in panel (b).

### F.2   Continuous Control Benchmark Tasks

We evaluate CQSM on continuous control benchmarks from the DeepMind Control Suite. DeepMind Control Suite [51] is a set of control tasks implemented in MuJoCo [50]. We choose TD3 [9], SAC [16], and Diffusion-QL [55] as three baselines for comparison. Below, we first provide a brief review of prior methods.

Policy gradient methods seek to directly optimize the policy by computing gradients of the expected reward with respect to the policy parameters [45]. Deterministic policy gradient algorithms for MDPs (with discrete time and continuous action space) are developed in [41] (DPG) and later extended to incorporate deep neural networks in [29] (DDPG). Recent studies have focused on stochastic policies with entropy regularization, also known as the softmax method; see for example, [31] (A3C); [39] (PPO).

- TD3: [9] proposed Twin Delayed Deep Deterministic Policy Gradient (TD3), which miti-gates overestimation bias in DDPG by using clipped double Q-learning and delayed policy updates. This yields more stable and accurate policy learning in continuous control tasks.

- SAC: The Soft Actor-Critic algorithm [16] optimizes a stochastic policy that maximizes both expected reward and policy entropy. The policy follows $\pi(a|x) \sim e^{\frac{1}{\lambda}Q(x,a)}$ and is reparameterized using a neural transformation of samples from a fixed distribution.

- Diffusion-QL: [55] integrates diffusion models with Q-learning by adding a term that maximizes action-values to the diffusion model's training loss. The final policy-learning objective is a linear combination of policy regularization and policy improvement.

We evaluate CQSM against the baselines (TD3, SAC, and Diffusion-QL) on a range of MuJoCo continuous control tasks, from high-dimensional domains (Cheetah Run, Walker Walk, Walker Run, Humanoid Walk) to simpler environments (Cartpole Balance, Cartpole Swingup). Each experiment is repeated with 10 random seeds, and we report the average episode return across runs (removing extremely poor results).

As shown in panels (a)-(d) of Figure F.2, CQSM matches or outperforms the TD3, SAC, and Diffusion-QL baselines, particularly in the early stages of training where it achieves higher rewards. This early performance gain highlights a key distinction between our approach and conventional actor-critic methods. Algorithms like SAC and TD3 may frequently sample actions with low Q-values in the initial stage because they do not utilize the action derivative $Q_a$, while Diffusion-QL can sometimes get stuck at suboptimal solutions. In contrast, our approach benefits from a more immediate policy adjustment via the learned score function, enabling the policy to better approximate the true optimal policy in the early stages. As a result, our method can achieve higher rewards in the early steps. In simpler tasks, shown in panels (e)-(f) of Figure F.2, CQSM achieves performance comparable to the baselines, demonstrating both its stability and general applicability across task complexities.

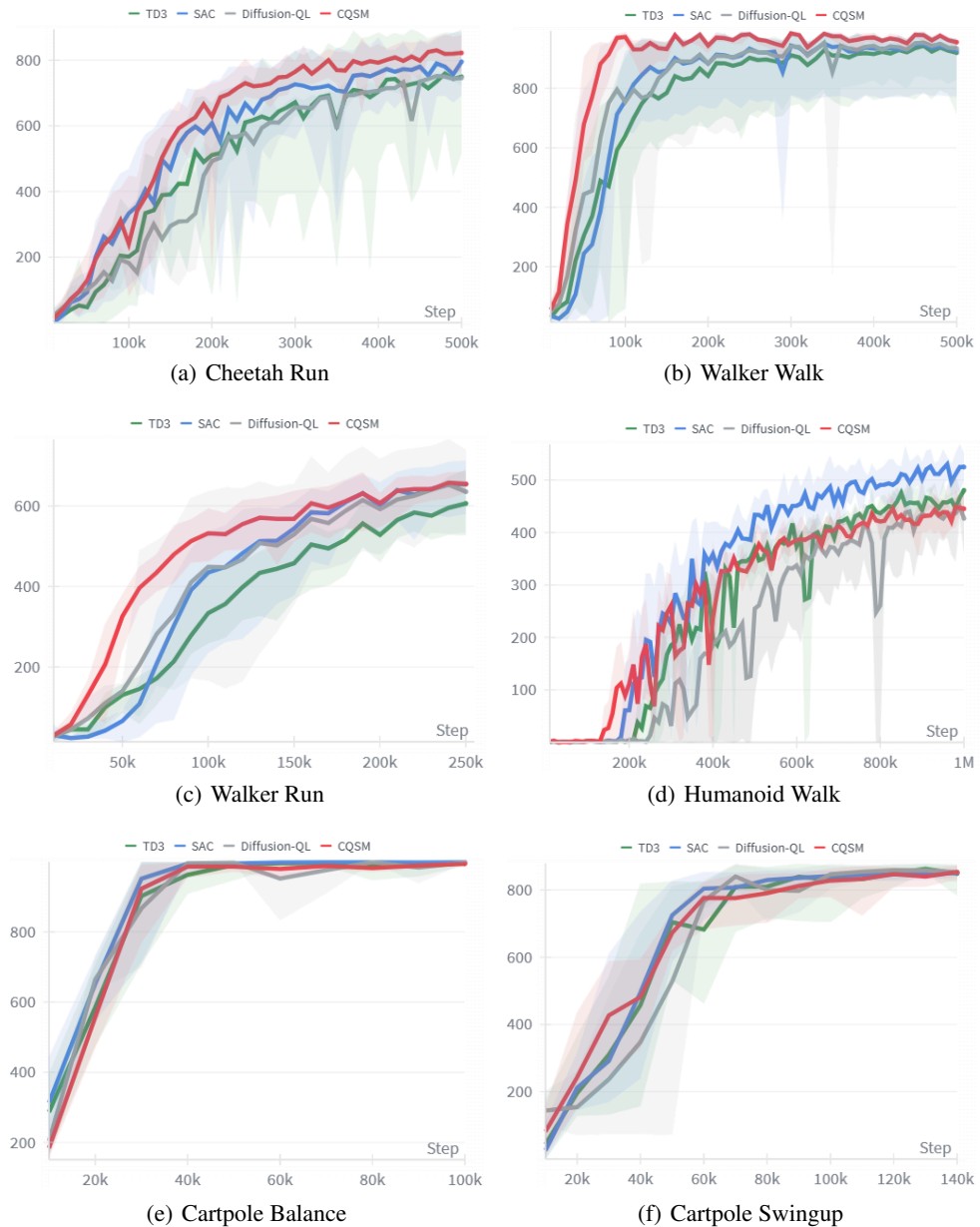

Figure F.2: Experimental Results Across A Suite of Six Continuous Control Tasks.

