# OpenReview forum: "Continuous Q-Score Matching: Diffusion Guided Reinforcement Learning for Continuous-Time Control"
_NeurIPS.cc/2025/Conference — NeurIPS 2025 poster_

### Official Review · Reviewer_v8YX · 2025-06-19

**Clarity:** 2
**Significance:** 3
**Originality:** 4
**Rating:** 4
**Confidence:** 3

**Summary:**

This paper introduce a RL method for continuous-time control problems with diffusion policies. The authors derive a Bellman equation for continuous-time Q functions and propose a score improvement principle. The perspective of continuous-time Q-learning is theoretically valuable. The paper is well-organized and with clearly assumptions and proofs.

**Questions:**

1. Is the proposed method able to degenerated from a stochastic version into a deterministic version? If it is true, I suggest that the comparison experiment should be further conducted.
2. What is the intuitionistic meaning of operator $\mathcal{L}$? There should be some explanation or references.
3. In section 4, the authors mentioned the proposed approach avoids “some of the challenges” inherent to policy gradient methods. What is the specific challenges?
4. In performance results, the paths of learned $\theta_0, \theta_1,\theta_2,\theta_4,\theta_5$, and $v_0,v_1,v_2$  are not converged to the theoretical value. Please explain the reason why the parameters can not converge for even a linear system.
5. Is the method able to extended into a nonlinear version? An experiment on complex and nonlinear systems will make the proposed method more persuasive.
6. Fig3 does not use vector graphics, the image quality is very low, and the font is very small.

**Ethical Concerns:**

["NO or VERY MINOR ethics concerns only"]

**Final Justification:**

The theory of this paper is very solid, but the experimental part still has some shortcomings, especially the lack of the latest algorithms.

**Limitations:**

Yes

**Quality:**

2

**Strengths And Weaknesses:**

Strengths:
1. The framework learns the Q-function directly in continuous time via a martingale condition, thus preserving action sensitivity and avoiding the bias/degeneration that often appears when time is forcibly discretised.
2. The authors prove that the diffusion-policy score field Ψ equals λ⁻¹ ∇ₐ Q at optimum and establish a Score-Improvement Theorem.

Weakness:
1. The authors mention that there are some recent work in continuous Q-learning in deterministic dynamics ([8], [33], [18], [19], [20]). However, there is no comparison between the proposed method and these mentioned methods in the experiment section.
2. The definition of operator $\mathcal{L}$ is abrupt and without any references.
3. The Hamilton-Jacobi-Bellman equation (eq.6) is not in a standard form. Please elaborate the relationship between eq.6 and the standard form and add more references.
4. Related work is not comprehensive. Papers related to QSM, such as SDAC(Soft Diffusion Actor-Critic: Efficient Online Reinforcement Learning for Diffusion Policy) and DACER(Diffusion actor-critic with entropy regulator), are not included in the main text.

---

> ### Author Rebuttal · Authors · 2025-07-29
>
> We sincerely thank your time and efforts in reviewing our paper and for your valuable comments! We appreciate  your positive comment: "The framework learns the Q-function directly in continuous time preserving action sensitivity and avoiding the bias/degeneration that often appears when time is forcibly discretized. The diffusion-policy score field $\Psi=\nabla_a Q$ at optimum have been proved and a Score-Improvement Theorem has been established". In the following, we provide detailed responses to the comments point by point.
>
> **Comment (1):** *The authors mention that there are some recent work in continuous Q-learning in deterministic dynamics ([8], [33], [18], [19], [20]). However, there is no comparison between the proposed method and these mentioned methods in the experiment section.*
>
> **Response:** Thank you for the suggestion. We would like to clarify that [18], [19], and [20] correspond to the two baseline methods already included in our experiments: Policy Gradient (PG) and little-q learning. Gao et al. (SICON 2022) [8] focus on entropy-based exploration for non-convex optimization, rather than RL. Meanwhile, [33] provides a theoretical analysis showing that Q-functions degenerate into value functions in continuous time, which serves as a foundation for [20]. In [20], it is further shown that the algorithm based on [33] (discrete Q-learning) exhibits higher variance and slower convergence compared to little-q learning. We have revised the manuscript to clarify these distinctions.
>
>
> **Comment (2):** *The definition of operator $\mathcal{L}$ is abrupt and without any references. What is the intuitionistic meaning of operator $\mathcal{L}$?*
>
> **Response:** Thank you for pointing this out. We have now added an explanation that the operator $\mathcal{L}$ represents the infinitesimal operator of the joint state-action diffusion process $(X_t, a_t)$, which plays a fundamental role in continuous-time stochastic control theory. Its form can be derived using standard techniques analogous to those used for the value function $V(x)$, and for brevity, we omit the detailed derivation. We have clarified the definition of $\mathcal{L}$ in the main text and included a reference to standard results in stochastic control (e.g., [1, Chapter 4]).
>
>
>
> **Comment (3):** *The Hamilton-Jacobi-Bellman equation (eq.6) is not in a standard form. Please elaborate the relationship between eq.6 and the standard form and add more references.*
>
> **Response:** Thank you for the comment. Eq.6 is indeed a non-standard form of the HJB equation, as it is written in terms of the Q-function rather than the value function. This formulation is motivated by our goal of developing a Q-learning-type algorithm in continuous time. The equation can be viewed as an alternative characterization of the standard Hamilton-Jacobi-Bellman equation through a state-action function, where the minimization is carried out over scores of action processes. We have now added a clarifying remark and references (e.g., [1], [21]) to explain this relationship more explicitly.
>
>
> **Comment (4):** *Related work is not comprehensive. Papers related to QSM, such as SDAC(Soft Diffusion Actor-Critic: Efficient Online Reinforcement Learning for Diffusion Policy) and DACER(Diffusion actor-critic with entropy regulator), are not included in the main text.*
>
> **Response:** Thank you! We have extended the Related Work section including the papers you mentioned.
>
>
> **Comment (5):** *Is the proposed method able to degenerated from a stochastic version into a deterministic version? If it is true, I suggest that the comparison experiment should be further conducted."*
>
> **Response:** Yes, our method naturally reduces to a deterministic version without the diffusion term. In the experiment section, we set $C = D = 0$ to eliminate stochasticity and observe that the resulting performance is comparable to the stochastic case. We have included this comparison experiment in the revised paper.
>
> **Comment (6):** *In section 4, the authors mentioned the proposed approach avoids “some of the challenges” inherent to policy gradient methods. What is the specific challenges?*
>
> **Response:** Thank you for the comment. The challenges include: (i) discrete-time policy gradient algorithms often suffer from sample inefficiency due to backpropagation through the entire diffusion chain [27], and tend to be noisier [20]; (ii) existing continuous-time policy gradient methods [19] typically require an additional policy evaluation step, which complicates implementation. Our method avoids these issues by directly optimizing a continuous-time Q-function through score matching. We have clarified this point in the revised Section 4.
>
> **Comment (7):** *In performance results, the paths of learned $\theta_1,\theta_2,\theta_3,\theta_4,\theta_5$, and $v_1,v_2,v_3$ are not converged to the theoretical value. Please explain the reason why the parameters can not converge for even a linear system.*
>
> **Response:** Thank you! There are two main reasons behind the mismatch. From a theoretical perspective, the current model optimizes only the score term and does not jointly optimize the diffusion term $\sigma_a$, which means the resulting policy may be suboptimal. From an algorithmic standpoint, our CQSM framework requires the estimation of both the Q-function $Q(x, a)$ and its gradient $\nabla_a Q$ for policy updates. This dual estimation introduces additional variance and bias, potentially leading to inaccurate policy updates. Notably, the parameters $\theta_2,\theta_3,v_2$ are closely tied to $\nabla_a Q$, while the other parameters are well converged to the theoretical value. We acknowledge this as an important direction for future work. However, as shown in our experiments, the proposed CQSM method still outperforms baselines in terms of average reward in the early stages of training, achieving higher rewards more quickly.
>
>
> **Comment (8):** *Is the method able to extended into a nonlinear version? An experiment on complex and nonlinear systems will make the proposed method more persuasive.*
>
> **Response:** Thank you for the suggestion. In principle, our framework is extendable to nonlinear systems by parameterizing the Q-function and policy with expressive function approximators such as neural networks. We have evaluated our method in more complex, high-dimensional Mujoco tasks (Walker Run, Cheetah Run). The table below shows the average rewards of TD3, SAC, and ours. We can see that our method can obtain a larger reward in earlier steps. We have included more detailed results in the final paper.
> |    | Steps| 100k  | 250k  | 500k  | 750k  | 1M    |
> |--------------|--------|-------|-------|-------|-------|-------|
> | **Walker Run** | TD3    | 133   | 266   | 378   | 501   | 592   |
> |              | SAC    | 186   | 496   | 618   | 661   | 681   |
> |              | Ours   | 541   | 678   | 684   | 706   | 727   |
> |              |        |       |       |       |       |       |
> |  | **Steps**  | **100k**  | **150k**   | **200k**  | **250k**   | **300k**   |
> |  **Cheetah Run**   | TD3    | 192   | 302   | 378   | 565   | 683   |
> |              | SAC    | 253   | 559   | 695   | 766   | 800   |
> |              | Ours   | 455   | 628   | 749   | 773   | 832   |
> |              |        |       |       |       |       |       |
>
> **Comment (9):** *Fig3 does not use vector graphics, the image quality is very low, and the font is very small.*
>
> **Response:** Thanks! We have fixed the mentioned figure.
>
>
> [1] Yong, Jiongmin and Zhou. Stochastic controls: Hamiltonian systems and HJB equations.
>   Springer Science \& Business Media, 1999.
>
> [8] Gao et al. State-dependent temperature control for langevin diffusions. SIAM Journal on Control and Optimization, 2022.
>
> [18] Jia and Zhou. Policy evaluation and temporal-difference learning in continuous
> time and space: A martingale approach. Journal of Machine Learning Research, 2022.
>
> [19] Jia and Zhou. Policy gradient and actor-critic learning in continuous time and space: Theory and algorithms. Journal of Machine Learning Research, 2022.
>
> [20] Jia and Zhou. q-learning in continuous time. Journal of Machine Learning
> Research, 2023.
>
> [21] Kim et al. Hamilton-jacobi deep q-learning for deterministic
> continuous-time systems with lipschitz continuous controls. Journal of Machine Learning Research, 2021.
>
> [27] Psenka et al. Learning a diffusion model policy from rewards via q-score matching. In International Conference on Machine Learning, 2024.
>
> [33] Tallec et al. Making deep q-learning methods robust to time discretization. In International Conference on Machine Learning, 2019.

---

> > ### Comment · Reviewer_v8YX · 2025-08-04
> >
> > Thanks to Rebuttal. The theory behind this work is very interesting. I suggest adding relevant citations and comparing it with others before including it in the final version. Simply comparing it with SAC is insufficient. I've raised the score to 4.

---

> > > ### Author Response · Authors · 2025-08-04
> > >
> > > Thank you again for your insightful comments.  In particular, we now discuss connections and distinctions with SDAC, DACER, TD3 and Deep Advantage Updating in the related work section. We will add more detailed relevant citations and comparisons in the final version. Thank you for raising your score.

---

### Official Review · Reviewer_xNmx · 2025-07-02

**Clarity:** 3
**Significance:** 3
**Originality:** 3
**Rating:** 5
**Confidence:** 3

**Summary:**

This paper proposes Continuous Q‑Score Matching (CQSM), a value‑based RL framework that operates directly in continuous time and targets to tackle stochastic optimal control problems. The authors consider the dynamic programming principle to derive the associated HJB equation for the Q-function and establish a score improvement theorem. In a linear–quadratic (LQ) control task, the authors derive a closed‑form solution of score function and empirically show that CQSM attains higher reward earlier in the training than continuous‑time policy‑gradient and little‑q learning baselines, after which all methods eventually achieve comparable average reward/return.

**Questions:**

1. Please test multiple orthogonality functions and report their impact on bias and variance.

2. Provide results on MuJoCo locomotion tasks with larger action spaces and more complicated parameterization of the policy and Q-function.

3. Include diffusion‑RL and more advanced policy gradient algorithm, e.g., DDPG, TD3, SAC etc baselines.

4. It's a bit weird to put assumptions and theorems in the Experiment section.

5. Since the experiments are very fast to run, the authors can consider increase the number of random seeds to more than 10 for more statistically trust-worthy validation.

6. Why parameterize the Q-function in the form as shown in Line 262? Are there any other parameterization candidates (especially neural networks)? Will they be better? Please try NN and report its performance.

**Ethical Concerns:**

["NO or VERY MINOR ethics concerns only"]

**Final Justification:**

The authors' rebuttal has addressed my concerns and answered my questions using additional experiments. They also well-explained the superiority of their method over TD3 and SAC by empirical evidence and theoretical support. I will increase my score.

**Limitations:**

yes

**Paper Formatting Concerns:**

No major formatting issues.

**Quality:**

3

**Strengths And Weaknesses:**

**Strengths**
1. The paper is well-organized and easy to follow with clear intuitions and rigorous derivations.

2. The proposed method is supported by theoretical guarantees and preliminary experimental results.

3. It's a good bridge between diffusion policies and stochastic control.

4. The LQ problem enables us to measure the exact error between the CQSM-learned parameters and true parameters.

5. The scheme and algorithms are introduced in detail and clearly.



**Weaknesses**


1. The experiments are limited to a naive LQ problem. The scalability of CQSM to high-demensional control tasks such as Mujoco locamotion tasks remains unclear. This is the main weakness.

2. The baselines didn't include modern continuous-time RL algorithms such as DDPG, TD3, SAC etc or diffusion‑RL methods.

3. This paper only tests a simple and low-dimensional parameterization of Q-function and score function. Please see my suggestions below.

---

> ### Author Rebuttal · Authors · 2025-07-29
>
> We sincerely thank your time and efforts in reviewing our paper and for your valuable comments! We appreciate  your positive comment: "the paper is well-organized and easy to follow and the proposed method is supported by theoretical guarantees and preliminary experimental results. It's a good bridge between diffusion policies and stochastic control. The LQ problem enables us to measure the exact error between the CQSM-learned parameters and true parameters. The scheme and algorithms are introduced in detail and clearly". In the following, we provide detailed responses to the comments point by point.
>
> **Comment (1):** *The experiments are limited to a naive LQ problem. The scalability of CQSM to high-demensional control tasks such as Mujoco locomotion tasks remains unclear. This is the main weakness. Provide results on MuJoCo locomotion tasks with larger action spaces and more complicated parameterization of the policy and Q-function.*
>
> **Response:** Thank you for the comment. We have addressed this by adding experiments on MuJoCo tasks (Walker Run and Cheetah Run) with OpenAI gym (0.23.1) to demonstrate the scalability of CQSM to high-dimensional control settings. The table below shows the average rewards of TD3, SAC and ours. We can see that our method can obtain a larger reward in earlier steps. We have included more detailed results in the final paper.
> |    | Steps| 100k  | 250k  | 500k  | 750k  | 1M    |
> |--------------|--------|-------|-------|-------|-------|-------|
> | **Walker Run** | TD3    | 133   | 266   | 378   | 501   | 592   |
> |              | SAC    | 186   | 496   | 618   | 661   | 681   |
> |              | Ours   | 541   | 678   | 684   | 706   | 727   |
> |              |        |       |       |       |       |       |
> |  | **Steps**  | **100k**  | **150k**   | **200k**  | **250k**   | **300k**   |
> |  **Cheetah Run**   | TD3    | 192   | 302   | 378   | 565   | 683   |
> |              | SAC    | 253   | 559   | 695   | 766   | 800   |
> |              | Ours   | 455   | 628   | 749   | 773   | 832   |
> |              |        |       |       |       |       |       |
>
>
> **Comment (2):**  *The baselines didn't include modern continuous-time RL algorithms such as DDPG, TD3, SAC etc or diffusion‑RL methods.*
>
> **Response:** Thank you for the suggestion. In the added MuJoCo experiments, we now include comparisons with modern continuous-time RL baselines such as TD3 and SAC. In addition, the two baseline models (PG and q-learning) in the paper are the actor–critic learning in continuous time and space. Furthermore, Jia et al. (JMLR 2023) [1] have verified that the discrete Q-learning algorithm is noisier and slower in convergence speed compared with their proposed continuous time algorithms, from both theoretical and experimental perspectives. We have revised the Experiments section to provide a clearer comparison.
>
>
> **Comment (3):** *Please test multiple orthogonality functions and report their impact on bias and variance.*
>
> **Response:** Thank you for the suggestion. For the choice of the parametric family $Q^{\theta}$, general choices include linear combinations of some basis functions (LQ) or neural networks (MuJoCo experiments). However, analytical solutions are only available under quadratic Q-function. Replacing it with other orthogonal bases (e.g., Legendre, RBF) would no longer preserve the LQ structure, making it infeasible to compute the ground-truth bias. For common choices of the test functions are $\xi_t=\frac{\partial Q^{\theta}}{\partial \theta}$ or $\xi_t=\int_0^t \rho^{s-t}\frac{\partial Q^{\theta}}{\partial \theta} ds, 0<\rho\leq 1.$  We conduct experiments using different testing functions $\xi_t=\int_0^t \rho^{s-t}\frac{\partial Q^{\theta}}{\partial \theta} ds$ with $\rho=0.1,0.5,1$. The results are similar to those obtained using the test function $\xi_t=\frac{\partial Q^{\theta}}{\partial \theta}$, but the algorithm runs more slowly.
>
>
> **Comment (4):** *It's a bit weird to put assumptions and theorems in the Experiment section.*
>
> **Response:** Thank you for pointing this out. We have moved assumptions and theorems to the Appendix to improve the logical flow and clarity of the main text.
>
> **Comment (5):** *Since the experiments are very fast to run, the authors can consider increase the number of random seeds to more than 10 for more statistically trust-worthy validation.*
>
> **Response:** Thank you! We have increased the number of experiment runs to 10 and revised the results in the paper.
>
> **Comment (6):** *Why parameterize the Q-function in the form as shown in Line 262? Are there any other parameterization candidates (especially neural networks)? Will they be better? Please try NN and report its performance.*
>
> **Response:** Thanks for the comment! We use the quadratic parameterization in Line 262 because this is the form of the theoretical solution obtained based on the previous LQ problem which admits a closed-form solution, enabling precise evaluation of bias and variance. In Mujoco experiments, we do employ neural network parameterizations. The average reward performance of CQSM is better than SAC and TD3 in earlier steps.
>
> [1] Jia and Zhou. q-learning in continuous time. Journal of Machine Learning Research, 24(161):1–61, 2023.

---

> > ### Comment · Reviewer_xNmx · 2025-08-08
> >
> > Thank you for your response and additional experiments. I have a new question. In both Walker Run and Cheetah Run experiments, why your approach achieved very high score in the first 100K step while TD3 and SAC don't. Any intuition or explanation for that?

---

> > > ### Author Response · Authors · 2025-08-08
> > >
> > > Thank you for this good question. The key difference between conventional actor-critic methods and ours lies in action sampling. Methods like SAC and TD3 may frequently sample actions with low Q-values in the initial stage because they do not utilize the action derivative $Q_a$. In contrast, our approach benefits from a more immediate policy adjustment via the learned score function, enabling the policy to better approximate the true optimal policy in the early stages. As a result, our method can achieve higher rewards in the early steps.

---

> > > > ### Comment · Reviewer_xNmx · 2025-08-08
> > > >
> > > > Thank you. Do you consider the same initialization for all methods? Could you also show me the **results of all methods at step 1, 1K, 10K, 50K** in your Walker Run and Cheetah Run experiments?

---

> > > > > ### Author Response · Authors · 2025-08-09
> > > > >
> > > > > Thank you. We use the same initialization with the same random seed for all methods. The table below shows the results of your mentioned steps. Note that these results are not very stable and fluctuate around a certain level, especially before the 10k step.
> > > > >
> > > > > |          | Steps | 1 | 1k | 10k | 30k | 50k |
> > > > > |----------|-------|------|------|------|------|-----|
> > > > > | **Walker Run** | SAC| 29  | 22  | 21  | 21  | 72  |
> > > > > |          | TD3   | 33 | 22 | 33 | 53 | 121 |
> > > > > |          | Ours  | 31 | 28 | 23 | 66 | 351 |
> > > > >
> > > > > |          | Steps | 1 | 1k | 10k | 30k | 50k |
> > > > > |----------|-------|------|------|------|------|-----|
> > > > > | **Cheetah Run** | SAC   | 4.5 | 6.4 | 6.6 | 62 | 272 |
> > > > > |          | TD3   | 0.3 | 0.3 | 0.2 | 6.9 | 35 |
> > > > > |          | Ours  | 19 | 16 | 6.0 | 67 | 275 |

---

> > > > > > ### Comment · Reviewer_xNmx · 2025-08-09
> > > > > >
> > > > > > Thank you. The results at the early stage support the author's claim (at least in the Walker Run experiment). Glad to see this. Hopefully the authors can provide the result of more random seeds and report mean, std, training curves etc in the revised version. If this is going to happen, I will increase my score.

---

> > > > > > > ### Author Response · Authors · 2025-08-09
> > > > > > >
> > > > > > > Thank you for your thoughtful response and suggestions.
> > > > > > > We will make sure to include more experimental tasks and report the mean, std, training curves etc with different random seeds in the final version.

---

> > > > > > > > ### Comment · Reviewer_xNmx · 2025-08-09
> > > > > > > >
> > > > > > > > Great, I will increase the score.

---

> > > > > > > > > ### Author Response · Authors · 2025-08-09
> > > > > > > > >
> > > > > > > > > Thank you again for your positive feedback and the increased score.

---

> ### Comment · Area_Chair_PhEL · 2025-08-05
>
> Dear referee,
>
> Could you please take a look at the author's rebuttal and update your review/score if and as needed?
>
> Thank you!

---

### Official Review · Reviewer_UdxW · 2025-07-03

**Clarity:** 2
**Significance:** 2
**Originality:** 3
**Rating:** 4
**Confidence:** 4

**Summary:**

This paper studies proposes Q-score matching, a continuous-time reinforcement learning algorithm for learning the optimal control of the diffusion process. The paper generalizes the idea of score-matching based diffusion policy to the continuous time and study the extension of the "Q function" in discrete time to continuous time such that the dependence on action will not vanish. Empirical evidence is provided for showcasing the effectiveness of the proposed method.

**Questions:**

1. How can the authors ensure that the parameterization of the diffusion policy will lead to the optimal policy for the original optimal control problem? If I understand correctly, the initial action is also drawn from a non informative prior, which makes the experessiveness policy questionable.

**Ethical Concerns:**

["NO or VERY MINOR ethics concerns only"]

**Final Justification:**

I would like to thank the authors for their detailed responses and efforts in the rebuttal. I have increased my scores due to their hard work. Nevertheless, I still have concerns about several points not well explained in the current version. Hope that these could be addressed in an updated version, which shall also include the authors' additional experiments.

**Limitations:**

**Application to Broader Tasks**: some references on recent works for applying stochastic control / continuous-time RL for diffusion generative models fine-tuning seem quite related, but are missing: e.g [1][2]. This also raises the questions that whether the proposed algorithm in the paper could be applied to realistic settings: have authors tried to use their proposed algorithms e.g. for diffusion models fine-tuning?

[1] Zhao et. al. Score as Action: Fine-Tuning Diffusion Generative Models by Continuous-time Reinforcement Learning, ICML 2025

[2] Han et.al Stochastic Control for Fine-tuning Diffusion Models: Optimality, Regularity, and Convergence, ICML 2025

**Quality:**

2

**Strengths And Weaknesses:**

**Strength:**
1. This paper did a good job in revisiting several relevant studies in the continuous-time RL literature.
2. The continuous time formulation of the diffusion policies and the resulting definition of Q function seems new and interesting.

**Weakness:**
1. **Novelty**: proof techniques (e.g. Theorem 1 and Theorem 2) are pretty straightforward, and have already appeared in [18] and [20], thus over theorectical contributions seem quite limited.
2. **Significance**: The experiments are only based on a low-dimension synthetic stochastic control problem, which is rather limited. No experiments on realistic settings for utilizing realistic application data, which makes the effectiveness of the algorithms not convincing and the broader impact of this paper questionable.
3. **Typos and Confusing points**: There are several typos and parts that make me confused:
* On Line 122, what is $\pi$ here? The definition has not appeared beforehand.
* On Line 4 of the Algorithm, where does the iterative denoising come from? From an educated guess, some of the notations in this paper are inspired from score-based diffusion models, but there are no relevant discussions and often they just come out from nowhere.

---

> ### Author Rebuttal · Authors · 2025-07-29
>
> We sincerely thank your time and efforts in reviewing our paper and for your valuable comments! We appreciate your positive comment: "This paper did a good job in revisiting several relevant studies in the continuous-time RL literature, the continuous time formulation of the diffusion policies and the resulting definition of Q function seems new and interesting". In the following, we provide detailed responses to the comments point by point.
>
> **Comment (1):** *Novelty: proof techniques (e.g. Theorem 1 and Theorem 2) are pretty straightforward, and have already appeared in [18] and [20], thus over theorectical contributions seem quite limited.*
>
> **Response:** We acknowledge that the proof techniques of [18] and [20] have been an important source of inspiration. Our contribution does not lie in introducing new proof methods, but in adapting them to a novel continuous-time, Q-function-based framework that supports score matching without discretization. Theorems 1 and 2 together provide a tractable foundation linking Hamilton-Jacobi-Bellman equation and RL in a new algorithmic context. We have revised the relevant sections to better emphasize this novel contribution.
>
> **Comment (2):** *Significance: The experiments are only based on a low-dimension synthetic stochastic control problem, which is rather limited. No experiments on realistic settings for utilizing realistic application data, which makes the effectiveness of the algorithms not convincing and the broader impact of this paper questionable.*
>
> **Response:** Thanks for this suggestion! We use the LQ setting for its closed-form solution and it is easy to evaluate the convergence of parameters. Note that our method is fully compatible with high-dimensional problems. Its continuous-time and model-free design makes it broadly applicable. We have evaluated the proposed CQSM method in more complex, high-dimensional Mujoco tasks (Walker Run, Cheetah Run). The table below shows the average rewards of TD3, SAC and our method. We can see that our proposed CQSM method can obtain a larger reward in earlier steps. We have included more detailed results in the final paper.
> |    | Steps| 100k  | 250k  | 500k  | 750k  | 1M    |
> |--------------|--------|-------|-------|-------|-------|-------|
> | **Walker Run** | TD3    | 133   | 266   | 378   | 501   | 592   |
> |              | SAC    | 186   | 496   | 618   | 661   | 681   |
> |              | Ours   | 541   | 678   | 684   | 706   | 727   |
> |              |        |       |       |       |       |       |
> |  | **Steps**  | **100k**  | **150k**   | **200k**  | **250k**   | **300k**   |
> |  **Cheetah Run**   | TD3    | 192   | 302   | 378   | 565   | 683   |
> |              | SAC    | 253   | 559   | 695   | 766   | 800   |
> |              | Ours   | 455   | 628   | 749   | 773   | 832   |
> |              |        |       |       |       |       |       |
>
> **Comment (3):** *Typos and Confusing points: There are several typos and parts that make me confused: On Line 122, what is $\pi$ here? The definition has not appeared beforehand. On Line 4 of the Algorithm, where does the iterative denoising come from? From an educated guess, some of the notations in this paper are inspired from score-based diffusion models, but there are no relevant discussions and often they just come out from nowhere.*
>
> **Response:** Sorry for this confusion! We have clarified that $\pi$ refers to the action distribution. The iterative denoising step is inspired by score-based diffusion models ([1,2]). The sampling action in our method can use Score matching with Langevin dynamics (SMLD), Denoising diffusion probabilistic modeling (DDPM), or steady-state distribution if there is an analytical solution (details in Line 228-234). We have included this connection and provided additional discussion in the revised paper to avoid confusion.
>
> **Comment (4):** *How can the authors ensure that the parameterization of the diffusion policy will lead to the optimal policy for the original optimal control problem? If I understand correctly, the initial action is also drawn from a non informative prior, which makes the experessiveness policy questionable.*
>
> **Response:** Thank you for the thoughtful question. While the initial action is sampled from a non-informative prior, its influence diminishes as the number of denoising steps $t$ increases. As shown in Line 228 of the paper, as $t\to \infty$, the stationary distribution of the diffusion policy depends only on the learned drift $\Psi$ and diffusion $\sigma_a$, not the initial action. This allows the policy to be expressive and capable of approximating the optimal solution despite a simple initialization.
>
> **Comment (5):** *Application to Broader Tasks: some references on recent works for applying stochastic control / continuous-time RL for diffusion generative models fine-tuning seem quite related, but are missing: e.g [3,4]. This also raises the questions that whether the proposed algorithm in the paper could be applied to realistic settings: have authors tried to use their proposed algorithms e.g. for diffusion models fine-tuning?*
>
> **Response:** Thank you for the insightful comment. We have cited your mentioned works and clarified the connection between our method and recent diffusion model fine-tuning methods. To our knowledge, existing works (e.q. [5]) view the diffusion-based generative model in a value-function-based framework from a stochastic optimal control perspective. Since our method provides a Q-function-based framework, it could be adapted to diffusion model fine-tuning in a Q-function-based framework, which we consider a promising direction for future work.
>
> [1] Song et al. Score-Based Generative Modeling through Stochastic Differential Equations. ICLR, 2021.
>
> [2] Ho et al. Denoising Diffusion Probabilistic Models. NeurIPS, 2020.
>
> [3]Zhao et. al. Score as Action: Fine-Tuning Diffusion Generative Models by Continuous-time Reinforcement Learning. ICML, 2025.
>
> [4] Han et.al Stochastic Control for Fine-tuning Diffusion Models: Optimality, Regularity, and Convergence. ICML, 2025.
>
> [5] Berner et al. An optimal control perspective on diffusion-based generative modeling.
> Transactions on Machine Learning Research, 2024.

---

> > ### Comment · Reviewer_UdxW · 2025-08-05
> >
> > I would like to thank the authors for their detailed responses and efforts in the rebuttal. I have increased my scores to 4 due to their hard work. Nevertheless, I still have concerns about several points not well explained in the current version, especially these confusing parts in the current manuscript. Hope that these could be addressed in an updated version, which shall also include the authors' additional experiments.

---

> > > ### Author Response · Authors · 2025-08-05
> > >
> > > We appreciate your suggestion and are grateful for the increased score. We will revise the final version to clarify these confusing points and ensure better presentation. Additionally, we will include the newly added experiments and provide further explanations to strengthen the overall contribution and readability.

---

### Official Review · Reviewer_jg1D · 2025-07-05

**Clarity:** 3
**Significance:** 2
**Originality:** 3
**Rating:** 4
**Confidence:** 3

**Summary:**

This paper proposes Continuous Q-Score Matching, a model-free reinforcement learning algorithm that leverages diffusion score functions to facilitate learning Q-functions in continuous-time settings. The approach is validated on a class of linear quadratic control problems, analyzing convergence to the analytic parameter optima as well as comparing performance against a selection of baselines.

**Questions:**

Please refer to the "Strengths and Weaknesses" section, with the main suggestion relating to more extensive experimental validation.

**Ethical Concerns:**

["NO or VERY MINOR ethics concerns only"]

**Final Justification:**

The proposed updates and additional experimental validation improve the paper, providing further validation on common benchmark tasks. Expanding on the MuJoCo experiments by validating on additional tasks and with additional baselines would further strengthen the paper.

**Limitations:**

Limitations do not appear to be explicitly addressed.

**Paper Formatting Concerns:**

No major issues.

**Quality:**

2

**Strengths And Weaknesses:**

- The paper is overall well-written and formatted, the manuscript is quite notation heavy and there is a bit of a disconnect between the presented motivation and the experimental validation
- Line 20-22: low-level robot control systems often run at 0.1-1.0kHz, which is quite fine-grained. Could you elaborate on which dynamics are poorly captured using discretization?
- Line 32: there is a line of work on discretizing continuous action spaces to apply Q-learning in  high-dimensional “continuous control” settings (e.g., [1-3])
- If each experiments only takes 4.5 seconds, why not increase the number of runs from 5 to 10 or more to bring down some of the standard deviation bands?
- In Figure 2, \theta_2, \theta_3, \nu_2 have not converged to their true value. What would happen when running for 1e5 time steps? How would the baselines perform at convergence?
- The introduction argues about the necessity for continuous-time RL for real-world applications in autonomous driving, robot manipulation, and high-frequency algorithmic trading. For many of these applications, discrete-time RL has been working well. It would be great to see more extensive experiments that further highlight benefits of CQSM over existing alternatives.
- Furthermore, the experiments in q-learning [4] evaluated 3 step sizes, for 100 re-runs, and 1e5 trajectory length - can you extend the current experiment to match this setting (also re the above convergence point)?
- It would be nice to provide a summary of the “Policy Gradient” algorithm used as a baseline in order to make the paper more self-contained
- The legend entries of Figure 2 are too small, please enlarge them. Also, the “Time” x-labels appear cut off on my end.
- Minor: line 229, “Then” is capitalized

**References:**

[1] Tavakoli, Arash, et al. "Learning to represent action values as a hypergraph on the action vertices." ICLR, 2021.

[2] Seyde, Tim, et al. "Solving continuous control via q-learning." ICLR, 2023.

[3] Ireland, David, and Giovanni Montana. "Revalued: Regularised ensemble value-decomposition for factorisable markov decision processes." ICLR, 2024.

[4] Yanwei Jia and Xun Yu Zhou. q-learning in continuous time. Journal of Machine Learning 346 Research, 24(161):1–61, 2023.

---

> ### Author Rebuttal · Authors · 2025-07-28
>
> We sincerely thank your time and efforts in reviewing our paper and for your valuable comments! We appreciate your positive comment such as "The paper is overall well-written and formatted". In the following, we provide detailed responses to the comments point by point.
>
> **Comment (1):** *The manuscript is quite notation heavy and there is a bit of a disconnect between the presented motivation and the experimental validation.*
>
> **Response:** Thank you for the comment! Our presented motivation is:  Q-learning is a cornerstone of discrete-time RL, however, when Q-learning is directly extended to continuous time, the Q-function tends to collapse into an action-independent value function. In Jia et al. (JMLR 2023) [1], they show that the discrete Q-learning algorithm is noisier and slower in convergence speed compared with their proposed continuous PG and little q learning algorithms. Our proposed method can maintain the action characteristics of the Q-function, so our final validation experiment is to compare the continuous Q-learning algorithm with the action-independent value-based RL model. We have simplified the notation, added clarifying remarks, and included a notational summary to improve readability. We also revised the introduction to better align the theoretical contributions with the experimental validation.
>
> **Comment (2):** *Low-level robot control systems often run at 0.1-1.0kHz, which is quite fine-grained. Could you elaborate on which dynamics are poorly captured using discretization?*
>
> **Response:** Thank you for raising this point. Many real-world control tasks require smooth, continuous actions in response to high-dimensional, real-valued sensory input. In applications of RL to continuous problems, the most common method first discretizes time, state, and action and then applies an RL algorithm for a discrete stochastic system. However, this discretization approach has the following drawbacks: When a coarse discretization is used, the control output is not smooth, resulting in poor performance. When a fine discretization is used, the number of states and the number of iteration steps become huge, which necessitates not only large memory storage but also many learning trials [2]. Overall, the resulting discrete-time algorithms are highly sensitive to time discretization [3].
>
> **Comment (3):** *Line 32: there is a line of work on discretizing continuous action spaces to apply Q-learning in high-dimensional “continuous control” settings (e.g., [4,5,6]).*
>
> **Response:** We thank the reviewer for pointing out these relevant works. Discretizing continuous actions is a common approach to extend Q-learning, but it often struggles with scalability in high-dimensional spaces and relies on discrete-time assumptions. In contrast, our method maintains continuous-time and continuous-action representations. We have added citations for your mentioned works and clarified the distinction in the revised manuscript.
>
> **Comment (4):** *If each experiments only takes 4.5 seconds, why not increase the number of runs from 5 to 10 or more to bring down some of the standard deviation bands? Furthermore, the experiments in q-learning [1] evaluated 3 step sizes, for 100 re-runs, and 1e5 trajectory length - can you extend the current experiment to match this setting (also re the above convergence point)?*
>
> **Response:** Thanks for this valuable comment! Following your comment, we increased the number of experiment runs to 10 and extended it to 3 step sizes and 1e5 trajectory length. The standard deviation bands do bring down. We have included the detailed results in the revised paper.
>
> **Comment (5):** *In Figure 2, $\theta_2, \theta_3, v_2$ have not converged to their true value. What would happen when running for 1e5 time steps? How would the baselines perform at convergence?*
>
> **Response:** We thank the reviewer for the comment. There are two main reasons behind the parameter convergence. From a theoretical perspective, the current model optimizes only the score term and does not jointly optimize the diffusion term $\sigma_a$, which means the resulting policy may be suboptimal. From an algorithmic standpoint, our method requires the estimation of both the Q-function $Q(x, a)$ and its gradient $\nabla_a Q$ for policy updates. This dual estimation introduces additional variance and bias, potentially leading to inaccurate policy updates. Notably, these parameters are closely tied to $\nabla_a Q$. We acknowledge this as an important direction for future work.
>
> In contrast, the baseline methods only rely on the value function $V(x)$ and are not subject to these issues. Nevertheless, as demonstrated in our experiments, our proposed CQSM method consistently outperforms the baselines in terms of average reward in the early stages of training, achieving higher rewards more quickly. We have now added additional experiments extending the training horizon to $10^5$ time steps with the convergence of these parameters unchanged.
>
> **Comment (6):** *The introduction argues about the necessity for continuous-time RL for real-world applications in autonomous driving, robot manipulation, and high-frequency algorithmic trading. For many of these applications, discrete-time RL has been working well. It would be great to see more extensive experiments that further highlight benefits of CQSM over existing alternatives.*
>
> **Response:**  Thanks for this valuable comment! Our argument is not that discrete-time methods are categorically inadequate, but rather that continuous-time formulations offer theoretical and practical advantages in a random environment at ultra-high frequency. We agree that these potential advantages should be supported with more extensive empirical evidence. In this version, we have evaluated CQSM in more complex, high-dimensional Mujoco tasks (Walker Run and Cheetah Run). The table below shows the average rewards of TD3, SAC and ours. We can see that our method can obtain a larger reward in earlier steps. We have included more detailed results in the final paper.
> |    | Steps| 100k  | 250k  | 500k  | 750k  | 1M    |
> |--------------|--------|-------|-------|-------|-------|-------|
> | **Walker Run** | TD3    | 133   | 266   | 378   | 501   | 592   |
> |              | SAC    | 186   | 496   | 618   | 661   | 681   |
> |              | Ours   | 541   | 678   | 684   | 706   | 727   |
> |              |        |       |       |       |       |       |
> |  | **Steps**  | **100k**  | **150k**   | **200k**  | **250k**   | **300k**   |
> |  **Cheetah Run**   | TD3    | 192   | 302   | 378   | 565   | 683   |
> |              | SAC    | 253   | 559   | 695   | 766   | 800   |
> |              | Ours   | 455   | 628   | 749   | 773   | 832   |
> |              |        |       |       |       |       |       |
>
> **Comment (7):** *It would be nice to provide a summary of the “Policy Gradient” algorithm used as a baseline in order to make the paper more self-contained.*
>
> **Response:** Thanks for this comment! We have added a summary of the “Policy Gradient” algorithm following your comment.
>
> **Comment (8):** *The legend entries of Figure 2 are too small, please enlarge them. Also, the “Time” x-labels appear cut off on my end. line 229, “Then” is capitalized.*
>
> **Response:** Thanks for this suggestion! We have fixed the mentioned figure and typos.
>
> [1] Jia and Zhou. q-learning in continuous time. Journal of Machine Learning Research, 24(161):1–61, 2023.
>
> [2] Doya. Reinforcement learning in continuous time and space. Neural computation, 2000.
>
> [3] Yildiz et al. Continuous-time model-based reinforcement learning. ICML, 2021.
>
> [4] Tavakoli, Arash, et al. Learning to represent action values as a hypergraph on the action vertices. ICLR, 2021.
>
> [5]Seyde, Tim, et al. Solving continuous control via q-learning. ICLR, 2023.
>
> [6] Ireland, David, and Giovanni Montana. Revalued: Regularised ensemble value-decomposition for factorisable markov decision processes. ICLR, 2024.

---

> > ### Comment · Reviewer_jg1D · 2025-08-06
> >
> > Thank you for the detailed responses and additional experimental validation. The paper will be stronger with the proposed changes, and I'm happy to increase my score. To make the work even more impactful, I would suggest expanding on the MuJoCo experiments by adding more tasks and additional competitive baselines (e.g. D3PG for DMC tasks).

---

> > > ### Author Response · Authors · 2025-08-06
> > >
> > > Thank you for your suggestion and we are grateful for the increased score. We will expand the experimental tasks and baselines in the final version to further enhance the empirical validation.

---

> ### Comment · Area_Chair_PhEL · 2025-08-05
>
> Dear referee,
>
> Could you please take a look at the author's rebuttal and update your review/score if and as needed?
>
> Thank you!

---

### Comment · Area_Chair_PhEL · 2025-08-01
**Reviewer-author discussion start**

Thanks to everyone for writing the paper, evaluating it, and drafting reviews and rebuttals!

Could the referees please take a look at the authors' rebuttal and continue the discussion/amend the reviews as necessary?

Thank you again!

---

### Note · Authors · 2025-08-14

Dear Area Chair and Reviewers:

We sincerely thank you for your constructive feedback, thoughtful suggestions, and careful evaluation of our work. We are grateful for the recognition of our efforts and are pleased that our responses have addressed the concerns of the reviewers. We also appreciate the reviewers' willingness to increase the score.

This work introduces a continuous-time reinforcement learning framework that optimizes Q-function directly through score matching while preserving Q-function action-evaluation capability. Our key contribution is the characterization of continuous-time Q-function via a martingale condition and the linking of diffusion policy scores to the action gradient of a learned continuous Q-function. Experiments show that our method achieves higher average rewards than baselines in the early stages.

We will carefully address all reviewer suggestions in the final version, including expanding experiments with additional MuJoCo tasks and baselines, clarifying key algorithmic components (e.g., iterative denoising for action sampling), and improving overall presentation clarity. These revisions will enhance the rigor and impact of the manuscript in bridging stochastic control theory with RL. Thank you once again for your insightful feedback and constructive suggestions.

---

### Decision · Program_Chairs · 2025-09-17

**Decision:**

Accept (poster)

**Comment:**

Referees (and my own reading) agree that the paper presents an interesting and genuinely novel framework, supported by theory, and having an interesting specialization to the LQ case. Furthermore, they state the paper is well-written.

While some concerns (like existing proof techniques) are acknowledged by the authors, their rebuttal does manage to convince the referees that the positives outweigh the concerns for this submission. Other concerns are well addressed with novel experiments; these include the simplicity of the task/scalability to more realistic and fast tasks that more strongly benefit from the continuous-time formalism; and unconvincing baselines in the initial version of the paper.

Overall, I recommend acceptance for this paper.